# Towards Overcoming Reasoning Shortcuts in Neurosymbolic Learning via Efficient Generative Proxies

**Panagiotis Lymperopoulos**                                                  *plympe01@tufts.edu*
*Department of Computer Science, Tufts University*

**Li-Ping Liu**                                                               *liping.liu@tufts.edu*
*Department of Computer Science, Tufts University*

**Reviewed on OpenReview:** *https://openreview.net/forum?id=Sl2aC9hiaN*

## Abstract

Symbol grounding, the task of linking high-dimensional sensory inputs to symbolic representations in neurosymbolic AI (NeSy), often suffers from *reasoning shortcuts*, where inputs are mapped to unintended concepts due to limited supervision. Reconstruction-based training can help mitigate these ambiguities, but its effectiveness depends strongly on the quality and capacity of the reconstruction model. In this work, we propose a new grounding framework, Efficient Generative Proxies (EGP), that cleanly integrates reconstruction-based training into a generative modeling perspective. EGP subsumes several existing grounding approaches as special cases. We further argue that the role of reconstruction should be to capture the underlying structure of the data rather than to faithfully reconstruct inputs. Accordingly, we design a reconstruction term that leverages the principle that similar inputs should correspond to similar concept labels, thereby substantially reducing grounding ambiguity. We also develop extensions that incorporate additional inductive biases through this reconstruction term, improving robustness in more complex tasks. We evaluate our approach on tasks susceptible to reasoning shortcuts from the *RSbench* benchmark, as well as on the multi-concept *ObjectMath* dataset, integrating EGP into state-of-the-art neurosymbolic learning frameworks. Experimental results demonstrate that EGP significantly improves grounding accuracy and effectively mitigates reasoning shortcuts across diverse settings. The code of EGP is available at https://github.com/tufts-ml/egp-towards-overcoming-reasoning-shortcuts.

## 1 Introduction

Neurosymbolic AI (NeSy) aims to integrate the adaptability of neural networks with the reasoning capabilities of symbolic logic. A common pattern in these systems is the use of neural models to ground symbolic reasoners on sub-symbolic inputs, such as pixels. This combination promises to improve interpretability and robustness across domains ranging from robotics to language-based reasoning (Yang et al., 2025; Bhuyan et al., 2024).

Despite these benefits, recent studies have revealed that NeSy grounding models are prone to *reasoning shortcuts* (Marconato et al., 2023; 2025), where the model learns unintended mappings that satisfy the symbolic constraints rather than capturing the correct semantics. For example, in a traffic scenario, a model might incorrectly identify a pedestrian as a red light because both imply the same "stop" action, thereby satisfying the logical knowledge while failing to learn the true visual concepts. Fundamentally, these shortcuts arise due to insufficient supervision, as the provided symbolic constraints often do not allow for identifying the unique, correct grounding among multiple valid solutions.

This failure mode is intrinsic to the standard discriminative training objective used in neurosymbolic learning. Typically, these methods maximize the likelihood of symbols that satisfy the constraint given the inputs.

When the supervision signal is ambiguous, letting multiple configurations of symbols satisfy the symbolic constraints, this objective only forces the model to assign probability mass to any valid assignment, regardless of its semantic correctness. Consequently, grounding models often converge to distributions that maximize this likelihood but are incorrect, such as collapsing inputs of different classes into the same symbol. These methods fail because they do not fully consider the intrinsic structure of the input data. We refer to these methods as lacking *perceptual grounding*, which ensures that the classifier's predictions are consistent with the structure of the input space. While adding a reconstruction-based objective can theoretically resolve these ambiguities by enforcing consistency between predicted symbols and the input, such methods have been shown unreliable in the neurosymbolic setting as reconstructing low-level details can obfuscate the signal relevant to the reasoning problem.

In this work, we introduce Efficient Generative Proxies (EGP), a framework aiming to mitigate reasoning shortcuts by considering neurosymbolic learning from a generative modeling perspective but without the burden of full generation. EGP enables models to exploit the structure of the input data to resolve ambiguity that can be distinguished from the data. Rather than reconstructing inputs pixel-by-pixel, EGP utilizes a lightweight approximation of the generative likelihood to enforce that similar inputs yield similar symbols and distinct inputs yield distinct symbols. We essentially use the reconstruction term as a handle to provide various inductive biases to the grounding model. We demonstrate that this approach significantly improves the grounding accuracy of state-of-the-art methods on *RSbench* benchmark (Bortolotti et al., 2024) tasks and the multi-concept *ObjectMath* dataset. We also include extensive experiments to validate optimization strategies for EGP and further explore its ability to resolve reasoning shortcuts in different tasks.

We summarize our contributions as follows: 1) Using a generative modeling perspective, we derive a framework for neurosymbolic grounding that encompasses current state-of-the-art methods as well as some reasoning shortcut mitigation strategies in one unified objective. 2) We use our framework to provide perceptual grounding to neurosymbolic learners without any additional supervision. 3) We demonstrate how to use EGP in multi-concept domains and how to inject additional inductive biases to grounding models without observing labeled samples. 4) We conduct extensive evaluations and show that EGP is highly effective at improving grounding accuracy across multiple methods and benchmarks.

## 2   Related Work

Neurosymbolic AI (NeSy) aims to integrate the adaptability of neural networks with the reasoning capabilities of symbolic logic. This combination promises to improve interpretability (Tziafas & Kasaei, 2022; Padalkar et al., 2023; Zhang & Sheng, 2024) and robustness (Michel-Delétie & Sarker, 2024; Goel et al., 2024a;b; 2022; Mao et al., 2025) in robotics (Capitanelli & Mastrogiovanni, 2024; Wang et al., 2024a), language-based reasoning (Quan et al., 2025; Fang et al., 2024; Yang et al., 2025; Li et al., 2020) and various application domains such as life sciences, biomedical applications and IoT (Fitas, 2025; Hoehndorf et al., 2025). A common pattern in neurosymbolic systems is using neural models for grounding symbolic reasoners on sub-symbolic inputs. Prominent frameworks include DeepProbLog (Manhaeve et al., 2018), which combines probabilistic logic programming with deep learning, SATnet (Wang et al., 2019), which introduces a satisfiability solver in the learning process, Logic Tensor Networks (Badreddine et al., 2022), which utilize fuzzy logic, ABL (Dai et al., 2019) which utilizes abductive reasoning and Semantic Loss (Xu et al., 2018), which enables neural models to learn to satisfy symbolic constraints on their output.

Grounding models often suffer from reasoning shortcuts (Marconato et al., 2023; 2025), where the model learns unintended mappings that satisfy the symbolic constraints rather than capturing the correct semantics. Reasoning shortcuts arise due to insufficient supervision (Marconato et al., 2023), as the symbolic constraints do not have sufficient information to identify the correct grounding.

In a typical approach to neurosymbolic grounding (Li et al., 2024; Lymperopoulos & Liu, 2025), input-symbol pairs are treated independently, and a classifier is learned to classify the input to the true symbol class. The classifier is trained to predict symbols that satisfy symbolic constraints. These methods need special training techniques to mitigate the issue of reasoning shortcuts. Then van Krieken et al. (2025b;a) show that reasoning shortcuts cannot be fully addressed under the independent assumption; they then propose to learn a proper distribution over valid grounding solutions but without identifying a single solution. Marconato et al. (2023)

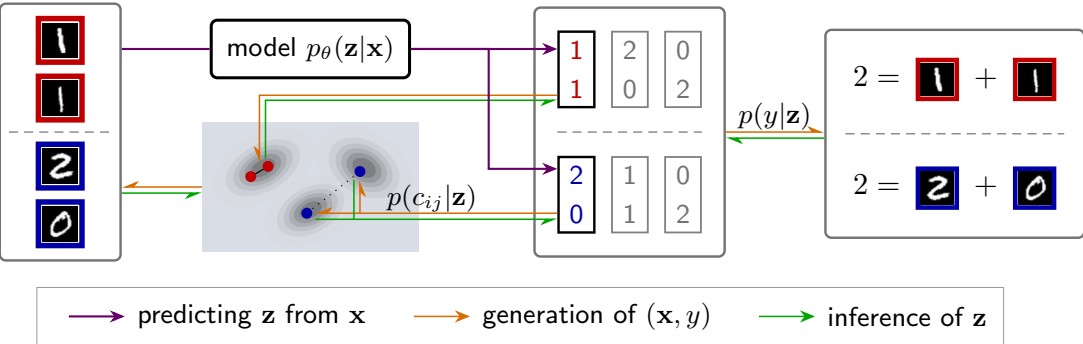

Figure 1: **Method Overview**. EGP is derived from a generative perspective on neurosymbolic learning that recovers common neurosymbolic approaches within it. The reasoning problem on the right cannot decide the exact labels of the MNIST digits involved. With the knowledge that two digits in the same (red) or different (blue) classes, their labels can be decided. EGP takes a generative approach (orange arrow): instance labels $\mathbf{z}$ decide the observed reasoning result $y$ and the relationship $c_{ij}$ of cluster memberships (in the same or different clusters). Supported by accurate inference (green arrow), the grounding model $p_\theta$ learns from inferred labels (purple arrow).

extensively studies strategies based on knowledge, data, or objectives for mitigating reasoning shortcuts. Particularly relevant is their experiment with reconstruction as an objective-based mitigation strategy, which yields no significant performance improvement, despite theoretical support. Another approach based on energy-based models (Sansone & Manhaeve, 2023) addressed some reasoning shortcuts, particularly ones related to degenerate solutions where all classes are collapsed into one. However, this approach still relies on the semantic loss and is unable to resolve reasoning shortcuts due to merging some clusters. Finally, recent work, Prototypical NeSy (Andolfi & Giunchiglia, 2025), resorts to prototype learning to avoid reasoning shortcuts. This approach creates a centroid/prototype for each class using labeled samples, and ensures the grounding model is consistent with those prototypes. While observing ground truth labels can definitively resolve ambiguity and improve performance, it is not always feasible to obtain such data.

The grounding with symbolic constraints can be viewed as a special case of weakly supervised learning (Zhou, 2018). The theory (Wang et al., 2020) indicates that the ambiguity of grounding labels cannot be avoided. Therefore, extra information or an inductive bias is needed to disambiguate these labels (Marconato et al., 2023). A related problem has been identified in concept bottleneck models (Bortolotti et al., 2025), which suffer from an analogous ambiguity when trained via weak supervision. These models typically differ from neurosymbolic approaches because the model predicting labels from symbols is learned. In this work, we focus on the neurosymbolic setting and show that the input reconstruction provides a convenient handle for introducing inductive bias to achieve perceptual grounding.

## 3 Methods

In a grounding problem, the input $\mathbf{x}$ consists of a sequence of $n$ observations, $\mathbf{x} = (\mathbf{x}_1, \ldots, \mathbf{x}_n)$, where each instance $\mathbf{x}_i \in \mathbb{R}^d$ is a high-dimensional input (e.g., an image or a bounding box). Each instance $\mathbf{x}_i$ is associated with an intrinsic but unobserved symbol $z_i \in \mathcal{Z}$ representing its concept or label. Let $\mathbf{z} = (z_1, \ldots, z_n)$ denote the vector of latent symbols. Given prior knowledge represented by the formula $\mathcal{K}$, an observed label $y$ is derived from the concepts $\mathbf{z}$ via a latent reasoning process. The relationship between the latent symbols $\mathbf{z}$ and the observed label $y$ is governed by the knowledge base $\mathcal{K}$, such that any valid pair $(\mathbf{z}, y)$ must satisfy the constraint $(\mathbf{z}, y) \vDash \mathcal{K}$. For example, if $\mathbf{z}$ represents the positions of objects on a map, $y$ might represent a successful plan that allows an agent to reach a goal while avoiding traps. We operate under the assumption that the knowledge base $\mathcal{K}$ is correct and known.

Our *goal* is to learn a neural grounding model $p_\theta(\mathbf{z}|\mathbf{x})$ from a dataset $\mathcal{D} = \{(\mathbf{x}^{(s)}, y^{(s)})\}_{s=1}^S$. Note that the number $n$ of input sequences and the constraint $\varphi$ may vary with problems in the dataset. We neglect the problem index $(s)$ in our discussion. If $y$ uniquely determines $\mathbf{z}$, then we can recover $\mathbf{z}$, possibly via an expensive computation, and the problem is fundamentally supervised learning. In reality, however, $y$ often cannot uniquely determine $\mathbf{z}$ but can only exclude illegal configurations of $\mathbf{z}$. In this case, we will aim to infer $\mathbf{z}$ from the joint information in both $y$ and $\mathbf{x}$ and then train the grounding model $p_\theta(\mathbf{z}|\mathbf{x})$.

The *reasoning shortcut* is a primary obstacle to training accurate grounding models. This phenomenon occurs when multiple latent explanations can account for a given reasoning label $y$, with the ground truth $\mathbf{z}$ being just one of many valid alternatives. Consequently, the grounding model faces an identifiability problem, as the training signal alone cannot distinguish the true explanation from spurious ones. For instance, as illustrated in Figure 1, if the sum of two MNIST digits is 2, there are three plausible labelings for the individual digits: (1, 1), (0, 2), or (2, 0). Additional benchmark examples, such as MNIST-Half, can be found in the work by Bortolotti et al. (2024). The primary goal of this work is to mitigate reasoning shortcuts by introducing a generative perspective that enhances the identification of the true latent explanation, thereby improving the accuracy of the grounding model.

### 3.1 Background

The discriminative approach learns $p_\theta(\mathbf{z}|\mathbf{x})$ by maximizing the conditional probability of $y$ given $\mathbf{x}$:

$$\log p_\theta(y|\mathbf{x}) = \max_\theta \log \sum_{\mathbf{z} \in \mathcal{Z}} p(y|\mathbf{z}) p_\theta(\mathbf{z}|\mathbf{x}). \tag{1}$$

Here $p(y|\mathbf{z}) = \mathbb{1}[\varphi(\mathbf{z}) = y]$ takes value 1 if $\mathbf{z}$ is consistent with $y$ or 0 otherwise. This formulation can be considered one type of weakly supervised learning and is equivalent to the semantic loss (Xu et al., 2018) .

Often the maximization problem is addressed by variational inference, where the true posterior $p(\mathbf{z}|y, \mathbf{x})$ is approximated by a variational distribution. For example, Li et al. (2024) and Lymperopoulos & Liu (2025) run MCMC sampling based predicted $\mathbf{z}$-s from $p_\theta(\mathbf{z}|\mathbf{x})$ to collect samples from the posterior. van Krieken et al. (2025a) maximizes the variational lower bound with a diffusion-based generative model.

Some work (Li et al., 2024; van Krieken et al., 2025b) uses a decomposable grounding model $p_\theta(\mathbf{z}|\mathbf{x}) = \prod_i p_\theta(z_i|\mathbf{x}_i)$, which assumes that the grounding model $p_\theta$ is a classifier and classifies each $\mathbf{x}_i$ individually. In this case, $p_\theta$ is often forced to learn the distribution of an individual $z_i$. The distribution is a marginalization of all ambiguous configurations of $\mathbf{z}$, thus its accuracy is often much lower than that trained with true labels.

### 3.2 Generative perspective of neurosymbolic learning

In our work, we learn a generative model $p(\mathbf{x}, y|\mathbf{z})$ from the data. This setup aligns with the assumed data generation procedure discussed by Marconato et al. (2023). In this approach, the model needs to consider the information about $\mathbf{x}$ when disambiguating $\mathbf{z}$. However, the major difficulty of this approach is that the generation of irrelevant details of $\mathbf{x}$ will obfuscate the inference about $\mathbf{z}$. In this work, we show that we can largely neglect the generation of low-level details and still learn a good grounding model.

Let the generative model be $p(\mathbf{x}, y, \mathbf{z}) = p(\mathbf{x}|\mathbf{z}) p(y|\mathbf{z}) p(\mathbf{z})$, where we assume that the target concept $\mathbf{z}$ generates observed inputs $\mathbf{x}$ and emits the weak label $y$. Then the log-likelihood of observed $\mathbf{x}$ and $y$ can be bounded with the ELBO:

$$\log p(y, \mathbf{x}) = \log \sum_{z \in \mathcal{Z}} p(y|\mathbf{z}) p(\mathbf{x}|\mathbf{z}) p(\mathbf{z}) \geq \mathbb{E}_{q(\mathbf{z}|\mathbf{x}, y)} \left[ \log p(\mathbf{x}, y, \mathbf{z}) - \log q(\mathbf{z}|\mathbf{x}, y) \right] \tag{2}$$

Here $q(\mathbf{z}|\mathbf{x}, y)$ is the variational distribution.

To train the grounding model in this generative setup, we put it into the variational distribution:

$$q_\theta(\mathbf{z}|\mathbf{x}, y) = \frac{q_\theta(\mathbf{z}|\mathbf{x}) p(y|\mathbf{z})}{Z_q}, \quad \text{where } Z_q = \sum_{\mathbf{z}' \in \mathcal{Z}} q_\theta(\mathbf{z}'|\mathbf{x}) p(y|\mathbf{z}') \tag{3}$$

Here $q_\theta(\mathbf{z}|\mathbf{x})$ is the grounding model. We use $q_\theta$ instead of $p_\theta$ to avoid confusion with the model distribution denoted by $p$. This specification of the variational distribution gives the variational lower bound:

$$\mathcal{L}_{\text{ELBO}} = \mathbb{E}_{q_\theta(\mathbf{z}|\mathbf{x},y)} \left[ \log \frac{\cancel{p(y|\mathbf{z})}p(\mathbf{x}|\mathbf{z})p(\mathbf{z})Z_q}{q_\theta(\mathbf{z}|\mathbf{x})\cancel{p(y|\mathbf{z})}} \right] \tag{4}$$

The rearranged lower bound is:

$$\mathcal{L}_{\text{ELBO}} = \underbrace{\log \left( \sum_{\mathbf{z}' \in \mathcal{Z}} q_\theta(\mathbf{z}'|\mathbf{x})p(y|\mathbf{z}') \right)}_{\text{Weak Supervision}} + \underbrace{\mathbb{E}_{q_\theta}[\log p_\theta(\mathbf{x}|\mathbf{z})]}_{\text{Reconstruction}} - \underbrace{\text{KL}[q_\theta(\mathbf{z}|\mathbf{x},y)||p(\mathbf{z})]}_{\text{Regularization}} \tag{5}$$

The first term is from the normalization term $Z_q$, which is taken out of the expectation. Note that this term is the same as (1). We set the prior $p(\mathbf{z})$ to be non-informative, so the third term is a constant.

Our formulation largely resembles objective-based mitigation of reasoning shortcuts by Marconato et al. (2023) but has a clear generative understanding. The reconstruction term encourages the grounding model $q_\theta(\mathbf{z}|\mathbf{x})$ to propose samples that are informative for reconstruction; therefore, the reconstruction term collects information from the input and helps $q_\theta(\mathbf{z}|\mathbf{x})$ learn the correct grounding function.

From the perspective of variational inference, the approach above minimizes $KL[q_\theta(\mathbf{z}|\mathbf{x})p(y|\mathbf{z})||p(\mathbf{z},y|\mathbf{x})]$. An alternative approach is training the grounding model $p_\theta(\mathbf{z}|\mathbf{x})$ using samples $\mathbf{z}$ from the true posterior $p(\mathbf{z}|\mathbf{x},y) \propto p(\mathbf{x}|\mathbf{z})p(y|\mathbf{z})p(\mathbf{z})$, then this alternative approach is equivalent to minimizing $KL[p(\mathbf{z},y|\mathbf{x})||q_\theta(\mathbf{z}|\mathbf{x})p(y|\mathbf{z})]$. Here, $p(y|\mathbf{z})$ is included for completeness. The two approaches have a similar learning objective of approximating the posterior, but the second approach is easier to understand: as long as $p(\mathbf{z},y|\mathbf{x})$ correctly samples correct $\mathbf{z}$ values, we can train a good grounding model $q_\theta(\mathbf{z}|\mathbf{x})$.

### 3.3 Reconstruction via cluster-based generative proxies

We show that there is substantial flexibility in designing the reconstruction term $p(\mathbf{x} \mid \mathbf{z})$ without explicitly modeling low-level details that are irrelevant to the target symbols. The key insight is that the generative model $p(\mathbf{x} \mid \mathbf{z})$ need not faithfully reconstruct $\mathbf{x}$, nor provide complete information about $\mathbf{z}$. Because $p(y \mid \mathbf{z})$ imposes strong constraints on $\mathbf{z}$, the reconstruction term only needs to provide complementary information that helps disambiguate label assignments consistent with $y$. Figure 1 summarizes our approach.

We design a reconstruction term in which $\mathbf{z}$ defines the cluster structure of $\mathbf{x}$, and this structure is then used to infer $\mathbf{z}$. Limiting the discussion to a single reasoning problem for simplicity of notation, we define $p(\mathbf{x} \mid \mathbf{z}) = p(\mathbf{x} \mid C)$, where $C \in \{0,1\}^{n \times n}$ indicates whether each pair of instances originates from the same latent class, $C_{ij} = \mathbb{1}[z_i = z_j]$. From a generative perspective, if $z_i = z_j$, then $\mathbf{x}_i$ and $\mathbf{x}_j$ are generated from the same mixture component; otherwise, they are generated from different components. Importantly, the class label $z_i$ itself does not specify which particular mixture component is used. Therefore, $C$ contains all necessary information about the cluster structure. The probability is essentially $p(\mathbf{x}_i, \mathbf{x}_j|C_{i,j})p(C_{ij}|z_i, z_j)$; compared with direct construction with $p(\mathbf{x}_i|\mathbf{z}_i)$, $C_{ij}$ blocks the information about low-level details, and thus the inference of $z_i$ and $z_j$ is less affected by factors irrelevant to the task (e.g. styles of MNIST digits).

In practice, we do not limit the approach to a single problem but compare instances within a random training batch. In this case, $C$ essentially contains the comparison between each pair of input instances in the dataset and thus provides rich information for the inference of $\mathbf{z}$. One alternative option is to use a full probabilistic approach without the pairwise approximation. In that case, we would assign a class label to each cluster according to the labels of instances in the cluster, allowing interactions between all instances simultaneously. However, this approach would require extra global parameters that must be inferred. By contrast, our pairwise approximation still captures the essence of distances between instances, retaining the capacity to disambiguate $z$ labels via their visual similarity while maintaining computational efficiency.

To stabilize training, we estimate the parameters associated with $p(\mathbf{z} \mid C)$ separately from the main objective. To capture the cluster structure of $\mathbf{x}$, we first map inputs into a low-dimensional embedding space and fit a Gaussian mixture model (GMM) with $|\mathcal{Z}|$ components in that space. The encoder can be trained using

unsupervised representation learning techniques (Bengio et al., 2013; Ericsson et al., 2022). Since the decoder is deterministic, evaluating $p(\mathbf{x} \mid C)$ reduces to computing the likelihood of compressed encodings under the fitted GMM. Conceptually, the neurosymbolic learning and the encoder have complementary roles: the encoder separates inputs by concept, but does not know the mapping between concepts and symbols. The neurosymbolic learning establishes that mapping.

We approximate the reconstruction term in two steps: 1) ignore the constraint imposed by $y$ inside the expectation; and 2) decompose the computation into instance pairs:

$$\mathbb{E}_{q_\theta(\mathbf{z}|\mathbf{x})p(y|\mathbf{z})} \left[p(\mathbf{x} \mid \mathbf{z})\right] \approx \sum_{\substack{i,j=1 \\ i \neq j}} \mathbb{E}_{q_\theta(\mathbf{z}|\mathbf{x})} \left[\log p(x_i, x_j \mid c_{ij})\right] \overset{\text{indp.}}{=} \sum_{\substack{i,j=1 \\ i \neq j}} \mathbb{E}_{q_\theta(z_i|\mathbf{x}_i)q_\theta(z_j|\mathbf{x}_j)} \left[\log p(x_i, x_j \mid c_{ij})\right]. \quad (6)$$

If we have the independence assumption, $q_\theta(\mathbf{z} \mid \mathbf{x}) = \prod_i q_\theta(z_i \mid \mathbf{x}_i)$, then the calculation in the last term has simpler calculation. In later training stages, most samples from $q_\theta$ satisfy the symbolic constraint imposed by $y$, so samples from $q_\theta$ are approximate samples from the true posterior. The exact computation of $p(\mathbf{x} \mid C)$ has all instances entangled and is hard to compute. The decomposition over instance pairs is a common approximation and greatly simplifies the computation.

With these approximations, the expectation admits a closed-form computation. The variational posterior of the consistency variable $C_{ij}$ is

$$q_\theta(C_{ij} = 1 \mid \mathbf{x}) = \sum_{\ell \in \mathcal{Z}} q_\theta(z_i = \ell \mid \mathbf{x}) q_\theta(z_j = \ell \mid \mathbf{x}) \overset{\text{indp.}}{=} \sum_{\ell \in \mathcal{Z}} q_\theta(z_i = \ell \mid \mathbf{x}_i) q_\theta(z_j = \ell \mid \mathbf{x}_j)$$

The independence assumption allows arbitrary pairing of instances, effectively augmenting training with diverse input pairs. Empirically, random instance pairs improve the performance of $q_\theta$.

Figure 1 illustrates the overall EGP framework. EGP integrates seamlessly with standard neurosymbolic learning while enforcing perceptual grounding by encouraging the model to respect input structure, without modeling low-level details that may obscure semantic information.

Although our reconstruction term does not attempt to recover $\mathbf{x}$, modeling relationships among $\mathbf{z}$ is often sufficient to disambiguate symbol assignments consistent with $y$. With the true posterior providing sufficient information about symbols $\mathbf{z}$, maximizing the resulting variational lower bound therefore yields a strong grounding model $q_\theta(\mathbf{z} \mid \mathbf{x})$. Our design also decouples clustering from the main training objective and helps stabilize the training procedure. An alternative would be to learn class-conditional generative components $p(\mathbf{x}_i \mid z_i = \ell)$ directly; however, this would require joint optimization of the generative and grounding models, increasing instability and complexity.

**Extension to multi-concept domains.** We now extend EGP to multi-concept settings, where each input $\mathbf{x}_i$ is associated with a vector of symbols $\mathbf{z}_i \in \mathcal{Z}^K$. Correspondingly, $C_{ij}$ becomes a $K$-dimensional vector $\mathbf{c}_{ij}$ where $c_{ij}^k = \mathbb{1}[z_i^k = z_j^k]$, which indicates a multi-view clustering structure of all $\mathbf{x}$. Our reconstruction term becomes:

$$\mathbb{E}_{q_\theta(\mathbf{z}|\mathbf{x})p(y|\mathbf{z})} \left[p(\mathbf{x} \mid \mathbf{z})\right] \approx \sum_{\substack{i,j=1 \\ i \neq j}} \sum_{k=1}^{K} \mathbb{E}_{q_\theta(z_i^k, z_j^k|\mathbf{x})} \left[\log p(\mathbf{x}_i, \mathbf{x}_j \mid c_{ij}^k)\right]. \quad (7)$$

To capture the multi-view clustering structure, the encoder can be learned using disentangled representation learning techniques (Wang et al., 2022; 2024b). More generally, domain knowledge can be injected into encoder training via specialized objectives or data augmentation. For example, in block-stacking tasks depending on shape and color, separate encoders can be trained for color (e.g., reconstructing color histograms) and shape (e.g., grayscale reconstruction). Alternatively, one may learn a split latent space with adversarial constraints to prevent information leakage across components.

## 3.4 Combining EGP with existing methods

The first term of Eq. (5) coincides with the conditional MLE objective in Eq. (1), widely used in neurosymbolic learning. State-of-the-art methods such as SoftenSG (Li et al., 2024), A2G (Lymperopoulos

& Liu, 2025), and NeSyDM (van Krieken et al., 2025a) optimize this objective using different techniques. Our framework recovers this objective within a broader formulation that additionally enforces perceptual grounding through the reconstruction term.

Different neurosymbolic methods can be instantiated in our framework in different ways. Methods assuming independent grounding (e.g., SoftenSG and A2G) can directly incorporate the reconstruction term during training. We refer to this approach as EGP-Direct. For methods such as NeSyDM, where computing the distribution $q_\phi(z_i|\mathbf{x}_i)$ is not possible and sampling is expensive, we directly derive weights of $\mathbf{z}$ samples from the cluster structure and use them to modify the label sampling procedure during weakly supervised learning, sampling labels that satisfy both the symbolic constraint and the EGP. Since the correct labels definitely respect the input structure, this makes them more likely to be discovered during sampling. We detail the modifications made to NeSyDM in the appendix.

### 3.5 Formal analysis of perceptual grounding

To theoretically quantify the benefit of our proposed method, we adopt the analysis framework of Marconato et al. (2023) and analyze the number of deterministic Reasoning Shortcuts (RSs) that minimize the training objective. Let $G$ denote the ground-truth concepts and $Z$ denote the learned symbols. A neurosymbolic predictor learns a distribution $p_\theta(Z|X)$, which in the deterministic limit implies a mapping $\alpha : G \to Z$. Standard neurosymbolic learning admits any map $\alpha$ that satisfies the logical constraints $\beta_K$.

Our method introduces a generative reconstruction term derived from unsupervised representations. To analyze its impact, we assume the unsupervised representation learning is sufficient to recover the equality of ground-truth concepts (Ideal Disentangled Clustering). Specifically, for a specific latent dimension or concept $k$, the generative model's posterior probability over the consistency variable $C_{ij}^{(k)}$ perfectly reflects ground-truth equality: $p(C_{ij}^{(k)} = 1 \mid x_i, x_j) \approx 1 \iff g_i^{(k)} = g_j^{(k)}$. Under this assumption, maximizing the EGP term acts as a hard constraint on the mapping $\alpha$, imposing an injectivity constraint on the component-wise mapping $\alpha_k$.

**Assumption 1.** *Marconato et al. (2023) Let $\mathbf{x}$ be a single input from a restricted space $\mathcal{X}$, and let $\mathcal{S}$ be a space containing auxiliary information. There exists an invertible function $f : \mathcal{S} \times \mathcal{Z} \to \mathcal{X}$ such that $\mathbf{x} = f(\mathbf{s}, z)$ for some $\mathbf{s} \in \mathcal{S}$, and $f$ is smooth in $\mathbf{s}$.*

**Assumption 2.** *Marconato et al. (2023) There exists a map $\beta_\mathcal{K}$ such that $y = \beta_\mathcal{K}(\mathbf{z})$.*

**Proposition 1** (Reduction of Reasoning Shortcuts via Perceptual Grounding)**.** *Consider a learning task with prior knowledge $K$ and logical label map $\beta_K$. Under Assumptions 1 & 2, and Ideal Disentangled Clustering, the number of deterministic optima is restricted to those maps that are both logically consistent and injective on the concept factors by:*

$$N_{opt} = \sum_{\alpha \in \mathcal{A}} \underbrace{\mathbb{I}\left\{ \bigwedge_{g \in supp(G)} \beta_\mathcal{K}(\alpha(g)) = \beta_\mathcal{K}(g) \right\}}_{Logical\ Consistency} \cdot \prod_{j=1}^{k} \underbrace{\mathbb{I}\left\{ \bigwedge_{\substack{g,g' \in supp(G) \\ g_j \neq g_j'}} \alpha_j(g_j) \neq \alpha_j(g_j') \right\}}_{Perceptual\ Grounding\ (Injectivity)} \tag{8}$$

Without perceptual grounding, the count is dominated by the Logical Consistency term, allowing for "many-to-one" mappings where distinct ground-truth concepts are collapsed into a single symbol. Our method introduces the second term, enforcing injectivity. Consequently, the search space is reduced from the set of all valid surjective maps to the set of valid permutations, which is significantly smaller. The proof is provided in the appendix. In practice it is possible that the clustering does not separate all concepts, in which case some incorrect mappings may still persist if allowed by the first term.

## 4 Experiments

In this section, we evaluate the proposed EGP method against multiple state-of-the-art neurosymbolic models across six tasks characterized by significant grounding ambiguity, including real world data and methods

| Method | $ACC_y$ | $ACC_\mathbf{z}$ | $ACC_{y,ood1}$ | $ACC_{\mathbf{z},ood1}$ |
|---|---|---|---|---|
| SoftenSG | $0.964 \pm 0.010$ | $0.437 \pm 0.003$ | $0.070 \pm 0.003$ | $0.398 \pm 0.002$ |
| SoftenSG+EGP | $0.960 \pm 0.009$ | $\mathbf{0.980 \pm 0.005}$ | $\mathbf{0.949 \pm 0.005}$ | $\mathbf{0.974 \pm 0.003}$ |
| A2G | $0.961 \pm 0.013$ | $0.440 \pm 0.005$ | $0.063 \pm 0.006$ | $0.395 \pm 0.007$ |
| A2G+EGP | $0.961 \pm 0.005$ | $\mathbf{0.978 \pm 0.004}$ | $\mathbf{0.950 \pm 0.009}$ | $\mathbf{0.977 \pm 0.005}$ |
| NeSyDM | $0.994 \pm 0.003$ | $0.717 \pm 0.008$ | $0.324 \pm 0.049$ | $0.645 \pm 0.031$ |
| NeSyDM+EGP | $0.991 \pm 0.003$ | $\mathbf{0.991 \pm 0.005}$ | $0.266 \pm 0.071$ | $\mathbf{0.707 \pm 0.027}$ |
| VAEL | $0.97 \pm 0.12$ | $0.45 \pm 0.11$ | $0.073 \pm 0.012$ | $0.442 \pm 0.022$ |
| GEDI | $0.423 \pm 0.012$ | $0.601 \pm 0.014$ | $0.319 \pm 0.020$ | $0.534 \pm 0.016$ |

Table 1: Effect of EGP on methods in the MNAdd-Half dataset. EGP completely eliminates reasoning shortcuts in this dataset and as a result consistently improves grounding performance across methods. Bold indicates statistical significance.

using additional supervision. In addition, we examine the two optimization strategies for Equation (6) when integrated with existing neurosymbolic methods. Finally, we perform a close comparison between EGP and pixel-level reconstruction (Marconato et al., 2023), demonstrating that the latter is ineffective in many cases despite directly optimizing the reconstruction loss.

### 4.1 Baselines

We consider three SOTA methods for neurosymbolic learning as baselines: SoftenSG (Li et al., 2024), A2G (Lymperopoulos & Liu, 2025), and NeSyDM (van Krieken et al., 2025a). SoftenSG and A2G both maximize the predictive probability in (1). NeSyDM employs a generative model to generate possible groundings $\mathbf{z}$, without much effort to disambiguate these groundings.

We combine EGP with the three methods and create three improved learning models: SoftenSG+EGP, A2G+EGP, and NeSyDM+EGP. We check the performance differences to measure the effectiveness of EGP. In the former two, which use the independence assumption, we optimize the approximate reconstruction term along with their main objectives. We use early stopping on the final label prediction accuracy over the validation set for those. For NeSyDM, we modify the training to favor grounding accuracy by using early stopping based on prediction stability on a validation set. This tends to improve grounding performance without observing concept labels, as it halts training after the model learns to satisfy the constraint but before it assigns probability to all possible incorrect solutions.

We also include two generative methods, VAEL, GEDI in the comparison. VAEL (Misino et al., 2022) is a generative method combining probabilistic reasoning with variational autoencoders. GEDI (Sansone & Manhaeve, 2023) is also a generative method but from a different perspective: it uses a contrastive term to approximate the density of the input data while learning the grounding with the standard semantic loss.

In addition, we also compare EGP against a prototype-based learning method (Andolfi & Giunchiglia, 2025). We use the name ProtoNeSy in our experiments to refer to the combination of DeepProbLog (Manhaeve et al., 2018) and the prototype method from the original paper. This method requires extra supervision from a few $\mathbf{z}$ labels to create prototypes for symbol grounding, so it often performs better than previous methods.

In our evaluation below, we run each experiment 4 times and report the mean and standard deviation for each metric. In all our experiments, to obtain compact encoders, we use N2D (McConville et al., 2019) trained only on samples from the training set. Additional implementation details and experiments with alternate encoders are available in the appendix. We evaluate performance in terms of grounding concept accuracy ($ACC_\mathbf{z}$) and label accuracy ($ACC_y$). We refer to performance in the out-of-distribution test sets as $ACC_{\mathbf{z},ood}$ and $ACC_{\mathbf{y},ood}$. We also report average F1 scores over $\mathbf{z}$ when necessary.

| Method | $ACC_y$ | $ACC_{\mathbf{z}}$ | $ACC_{y,ood1}$ | $ACC_{\mathbf{z},ood1}$ | $ACC_{y,ood2}$ | $ACC_{\mathbf{z},ood2}$ |
|---|---|---|---|---|---|---|
| SoftenSG | **0.920 ± 0.012** | 0.014 ± 0.002 | 0.043 ± 0.002 | 0.019 ± 0.005 | 0.007 ± 0.005 | 0.045 ± 0.016 |
| SoftenSG+EGP | 0.721 ± 0.057 | **0.456 ± 0.011** | **0.232 ± 0.009** | **0.463 ± 0.013** | **0.459 ± 0.139** | **0.539 ± 0.126** |
| A2G | **0.918 ± 0.009** | 0.013 ± 0.005 | 0.046 ± 0.003 | 0.016 ± 0.004 | 0.009 ± 0.006 | 0.043 ± 0.017 |
| A2G+EGP | 0.715 ± 0.044 | **0.461 ± 0.017** | **0.233 ± 0.011** | **0.459 ± 0.015** | **0.453 ± 0.122** | **0.576 ± 0.140** |
| NeSyDM | **0.958 ± 0.036** | 0.156 ± 0.025 | 0.000 ± 0.000 | 0.141 ± 0.017 | 0.002 ± 0.003 | 0.082 ± 0.031 |
| NeSyDM+EGP | 0.910 ± 0.052 | **0.455 ± 0.058** | **0.013 ± 0.014** | **0.313 ± 0.050** | **0.074 ± 0.050** | **0.364 ± 0.087** |
| VAEL | 0.961 ± 0.090 | 0.073 ± 0.062 | 0.038 ± 0.010 | 0.080 ± 0.013 | 0.060 ± 0.013 | 0.077 ± 0.059 |
| GEDI | 0.132 ± 0.010 | 0.344 ± 0.021 | 0.139 ± 0.014 | 0.330 ± 0.019 | 0.214 ± 0.024 | 0.353 ± 0.015 |

Table 2: Effect of EGP on methods in the *MNAdd-EvenOdd* dataset. EGP consistently improves grounding performance across methods, but final prediction performance drops and grounding accuracy still remains relatively low. This is because the dataset by construction is highly ambiguous, even allowing a complete permutation of concept labels while still achieving high label prediction accuracy. EGP settles at a local minimum due to extreme ambiguity. Bold indicates statistical significance.

| Method | $ACC_y$ | $ACC_{\mathbf{z}}$ | $ACC_{y,ood1}$ | $ACC_{\mathbf{z},ood1}$ |
|---|---|---|---|---|
| SoftenSG | 0.625 ± 0.217 | 0.375 ± 0.361 | 0.750 ± 0.144 | 0.750 ± 0.144 |
| SoftenSG+EGP | **1.000 ± 0.000** | **1.000 ± 0.000** | **1.000 ± 0.000** | **1.000 ± 0.000** |
| A2G | 0.462 ± 0.091 | 0.580 ± 0.316 | 0.383 ± 0.169 | 0.365 ± 0.185 |
| A2G+EGP | **1.000 ± 0.000** | **1.000 ± 0.000** | **1.000 ± 0.000** | **1.000 ± 0.000** |
| NeSyDM | 0.493 ± 0.022 | 0.679 ± 0.034 | **0.521 ± 0.132** | 0.363 ± 0.052 |
| NeSyDM+EGP | **0.698 ± 0.018** | **0.813 ± 0.021** | 0.091 ± 0.040 | **0.707 ± 0.034** |
| VAEL | 1.000 ± 0.000 | 0.996 ± 0.004 | 1.000 ± 0.000 | 0.995 ± 0.005 |
| GEDI | 1.000 ± 0.000 | 0.994 ± 0.002 | 1.000± 0.000 | 0.993 ± 0.003 |

Table 3: Effect of EGP on methods in the *MNLogic* dataset. As in *MNAdd-Half*, EGP completely eliminates reasoning shortcuts in this dataset and as a result consistently improves performance across methods. Bold indicates statistical significance. Other generative methods (VAEL,GEDI) also mitigate the ambiguity in this dataset.

## 4.2 Grounding hand-written digits in MNIST reasoning tasks.

In this experiment, we evaluate EGP and baseline methods on three MNIST reasoning tasks, *MNAdd-Half*, *MNAdd-EvenOdd*, and *MN-Logic* from *RSbench* (Bortolotti et al., 2024). The datasets are constructed such that multiple incorrect classifiers can achieve high accuracy on the final label $y$. *MNAdd-Half* and *MNAdd-EvenOdd* consist of pairs of handwritten digits and their sum as the supervision. The digit pairs in each dataset are chosen so that there is significant ambiguity, as multiple digit mappings can satisfy the data. The datasets also include out of distribution test sets consisting of digit combinations unseen during training. In *MN-Logic*, samples consist of MNIST digits from classes 0 and 1, combined in random 3-bit XOR formulas and their result. All these datasets also include additional out-of-distribution test-sets consisting of equations of the same digits, but in different digit combinations.

Table 1 shows results across methods in the *MNAdd-Half* dataset. In this dataset, a model can merge digit classes while still satisfying all sums. EGP eliminates these ambiguities by forcing the classifier to respect the input structure and thus significantly improves performance across methods. VAEL uses a decoder but fails to resolve the ambiguity because it focuses on low-level pixel details. GEDI mitigates some ambiguity by preventing extremely degenerate solutions, but in the NeSy setting it still relies on the semantic loss to satisfy constraints and does not address the consistency between choices of $\mathbf{z}$ and $\mathbf{x}$. NeSyDM generalizes poorly to unseen digit combinations as it models the joint probability of symbols and thus is less able to generalize to new symbol combinations. EGP provides no benefit in that case as the sampling-based strategy does not eliminate all incorrect symbolic solutions in this dataset. We further explore the differences between EGP implementations in 4.6.

Table 2 shows methods' performance in the *MNAdd-EvenOdd* dataset. This dataset admits a large number of ambiguous solutions. Most notably, it is possible to achieve perfect label performance by permuting labels between even and odd digits (e.g. 0 to 5, 1 to 6, etc). It is easy to conclude that it is not possible to learn

the correct grounding model based on the observed data. Baseline methods all manage to predict the correct sum with high accuracy but fail to correctly ground most samples. Across methods, EGP enables models to select the correct label about half the time, significantly improving performance. However, in this dataset, label prediction performance with EGP is lower. This is due to the excessive ambiguity of this dataset that EGP alone cannot resolve.

Taking a closer look at the trained models, we observe that the inferred label grounding accuracy during training converges to around 50% in both methods. Figure 2 shows the confusion matrix for a run of SoftenSG+EGP in this dataset. Despite the inductive bias provided by EGP, there is still too much ambiguity in the labels. While theoretically the model could choose one of the two label permutations and satisfy both the symbolic constraint and the EGP, instead it seems to settle at local minima where the clusters merged correspond to the label permutation that satisfies the symbolic constraints. The label prediction accuracy is then lower as the model tries to escape those minima unsuccessfully. While EGP seems to essentially eliminate all other incorrect groundings, it fails to prevent this one since the ambiguity cannot be fully resolved even with the exploitation of the clustering structure.

Table 3 shows results for the *MNLogic* Dataset. This dataset is simple in construction, as it only involves digits from the classes 0 and 1 of the MNIST dataset. Still, discriminative baseline methods struggle to learn the correct mapping, instead often merging the classes. EGP enables SoftenSG and A2G to perfectly solve the task, while it moderately improves the performance of NeSyDM, following the trend in previous datasets.

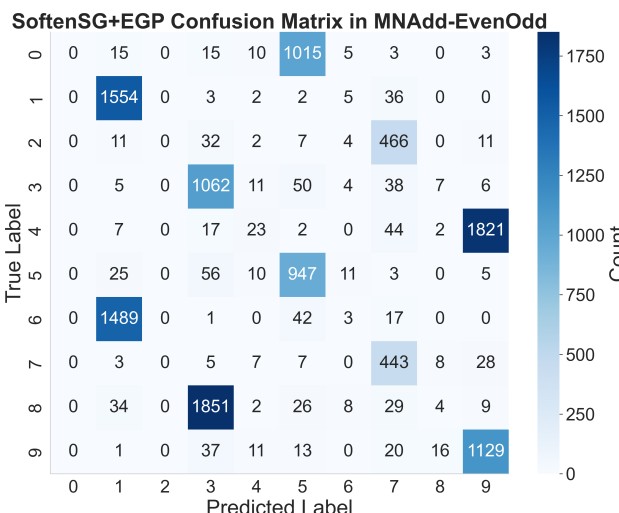

Figure 2: SoftenSG+EGP confusion matrix in *MNAdd-EvenOdd*. EGP eliminates almost all reasoning shortcuts but due to the extreme ambiguity of the task, where full label permutations are allowed, the model converges to a local optimum where the permuted classes are merged. When observing even a single equation using the merged classes, EGP resolves the ambiguity.

This is due to the implementation of EGP on NeSyDM, which modifies the label sampling procedure to sample labels that minimize the EGP term for each observed equation independently. This provides less rich feedback to the model than the alternate implementation methods that can be easily applied across any two inputs in the dataset. While it may be possible to further modify the sampling procedure to consider the EGP between samples in a batch, we leave that investigation to future work. VAEL and GEDI also perform well in this task. VAEL is able to overcome the ambiguity likely because there are only two visually very distinct classes (0 and 1) in the data, making pixel reconstruction effective in this setting. GEDI succeeds likely because it prevents full representation collapse, which in this dataset happens to correspond to the only ambiguous solution.

## 4.3 Grounding object properties in reasoning tasks with ObjectMath objects

In this experiment we evaluate the multi-concept version of EGP in the *ObjectMath* dataset. This dataset consists of images of 3 objects, where each object has one of three colors, one of two textures and one of three shapes. The reasoning problem is another arithmetic problem where each possible shape and color is assigned a number and the final result is computed by multiplying the attributes of each object and summing up the results. For each scene we observe the computation result as supervision. Due to commutativity of addition, and due to multiple attribute combinations resulting in the same result, the task is highly ambiguous. Figure 3 left shows an illustrative example of the dataset.

To apply neurosymbolic learning with the independence assumption in this problem, we first run an object detection workflow to isolate images of objects in the scene. We implement 2 versions of EGP and apply them to baselines to validate the efficacy of EGP in this domain. The first version uses a split latent space

| Metric | SoftenSG | SoftenSG+EGP | SoftenSG+EGP* | A2G | A2G+EGP | VAEL | GEDI |
|---|---|---|---|---|---|---|---|
| $ACC_y$ | $0.858 \pm 0.120$ | $0.715 \pm 0.042$ | $0.988 \pm 0.005$ | $0.855 \pm 0.115$ | $0.720 \pm 0.045$ | $0.860 \pm 0.040$ | $0.810 \pm 0.019$ |
| $ACC_{\mathbf{z}}$ | $0.602 \pm 0.091$ | $0.908 \pm 0.010$ | $0.970 \pm 0.016$ | $0.610 \pm 0.085$ | $0.915 \pm 0.012$ | $0.646 \pm 0.076$ | $0.685 \pm 0.103$ |

Table 4: SoftenSG variants in the multi-attribute *ObjectMath* dataset. SoftenSG+EGP* achieves near perfect performance as it leverages domain knowledge about concept-relevant features. SoftenSG+EGP achieves very high concept accuracy compared to the baseline method, but lower label prediction as the fully unsupervised multi-attribute EGP may not always capture the desired concept-related features.

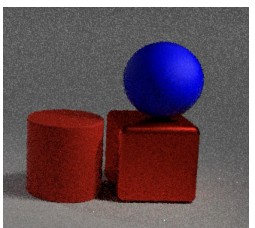 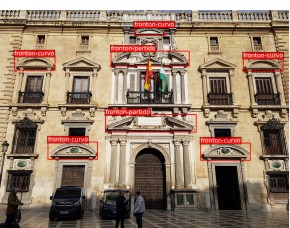

Figure 3: Illustrative examples for ObjectMath and MonumAI datasets. Left: synthetic Object-Math scene. Right: MonumAI facade example with outlined architectural elements.

| Method | Avg. $\mathbf{z}$ F1 $\uparrow$ | Label $y$ F1 $\uparrow$ |
|---|---|---|
| SoftenSG | $57.32 \pm 1.21$ | $92.11 \pm 2.41$ |
| SoftenSG+R | $56.87 \pm 1.32$ | $91.74 \pm 2.39$ |
| SoftenSG+EGP | $\mathbf{90.26 \pm 3.22}$ | $\mathbf{93.67 \pm 2.15}$ |
| A2G | $55.41 \pm 2.03$ | $91.26 \pm 2.13$ |
| A2G+EGP | $\mathbf{91.54 \pm 4.01}$ | $\mathbf{94.22 \pm 2.83}$ |
| VAEL | $65.64 \pm 3.22$ | $96.75 \pm 5.31$ |
| GEDI | $59.91 \pm 1.31$ | $90.81 \pm 2.13$ |

Table 5: Results on the MonumAI dataset. EGP significantly outperforms the baseline methods in a realistic setting with complex visual inputs.

autoencoder with a disentanglement objective to learn two distinct latent spaces capturing important features of each object. In the second version, EGP*, we inject additional inductive biases in the form of specially trained encoders that capture the two relevant features of each object, namely

color and shape. We then apply multi-concept EGP as normal in both methods. We use the EGP-Direct variant in both cases. Additional implementation details are available in the appendix.

Table 4 shows the results of this experiment. Both versions of EGP significantly boost grounding performance over the baseline, with the EGP* method achieving near-perfect performance. The label prediction accuracy of SoftenSG+EGP however is lower than the baseline approach. Examining the learned encoding more closely, one of the latent spaces tightly clusters objects around color, however the other space seems to separate spheres from the other shapes but not cylinders and cubes as effectively. As a result, EGP is uninformative in distinguishing the two shapes, providing less information to the classifier. Still, grounding accuracy is significantly higher than the baseline, making the trade-off favorable when grounding accuracy is primarily of interest. As a result, while EGP is effective when representation learning can disentangle features, injecting application-specific inductive biases in EGP when possible can yield even better performance. VAEL and GEDI fail to mitigate ambiguity in the real-world data.

## 4.4 Grounding elements of architecture styles in the MonumAI dataset

To validate EGP in more realistic settings where visual inputs are complex, we use the MonumAI dataset (Lamas et al., 2021). We construct a NeSy task by creating a rule set to map buildings into architectural styles from their architectural elements. We filter out samples that do not conform to the rules. These rules are ambiguous, providing weak supervision, but they preserve the correct building mapping in the filtered dataset. Figure 3 shows an illustrative example of the dataset.

The dataset provides bounding boxes of architectural elements. In the NeSy task, we are given a sequence of architectural elements and a style label for the building they belong to. The model is trained to predict the element label ($\mathbf{z}$) for each element. All supervision information comes from the style label ($y$) and the rule set. We evaluate SoftenSG, A2G variants and VAEL and GEDI. We also include SoftenSG+R (which reconstructs pixels with a decoder). For EGP, we use pretrained ResNet features for the encoder. Due to class imbalance, we report the F1 score.

| Metric | VAEL | GEDI | SoftenSG | SoftenSG+EGP | A2G | A2G+EGP | ProtoNeSy |
|---|---|---|---|---|---|---|---|
| $ACC_{\mathbf{z}}$ | $0.343 \pm 0.045$ | $0.327 \pm 0.082$ | $0.375 \pm 0.029$ | $0.948 \pm 0.014$ | $0.359 \pm 0.032$ | $0.945 \pm 0.015$ | $0.940 \pm 0.000$ |
| Avg. $F1_{\mathbf{z}}$ | $0.227 \pm 0.081$ | $0.282 \pm 0.081$ | $0.199 \pm 0.043$ | $0.942 \pm 0.012$ | $0.195 \pm 0.048$ | $0.940 \pm 0.014$ | $0.940 \pm 0.000$ |

Table 6: Performance comparison of baseline and EGP variants with ProtoNeSy on Kand-Logic. The first six methods use a few samples with $y$ labels from different knowledge $\mathcal{K}$ as additional supervision. ProtoNeSy uses a few samples with $\mathbf{z}$ labels as additional supervision.

The results in Table 5 demonstrate that EGP significantly outperforms the baseline method. Even though the visual input is much more complex than MNIST digits or objects in the ObjectMath dataset, EGP effectively leverages the internal data structure to disambiguate the latent concepts. In contrast, pixel-level reconstruction (SoftenSG+R) provides no significant benefit over the baseline, further highlighting the advantage of our cluster-based generative proxy.

## 4.5   Grounding with extra supervision information

In this subsection, we compare EGP against ProtoNeSy, which requires extra labeled $\mathbf{x}$ as prototypes in different $z$ classes. This is especially useful in highly-ambiguous settings such as the *MNAdd-EvenOdd* and *Kand-Logic* datasets from RsBench. The prototype-based method uses $\mathbf{z}$ labels to make these tasks learnable. In contrast, we provide other baselines and EGP variants with a weaker form of additional supervision for comparison.

In the *MNAdd-EvenOdd* dataset we add one sample equation from each digit combination (0,5), (1,6), (2,7), (3,8) and (4,9) to the training set. We leave the validation and test sets the same. Crucially, these equations do not resolve the global permutation on their own, as a model can still satisfy all equations by inverting these pairs of symbols. Still, the added constraints imposed by these equations, namely penalizing cases where the model merges, rather than swaps, these clusters, is sufficient for EGP to take effect and resolve the ambiguity. In this task, SoftenSG+EGP achieves $ACC_{\mathbf{z}}$ of $0.913 \pm 0.004$ and $ACC_y$ of $0.833 \pm 0.013$. This performance is slightly worse than the performance of ProtoNeSy ($ACC_{\mathbf{z}}$ of $0.97 \pm 0.03$), but it is achieved with a more convenient supervision form. Rather than directly labeling the visual inputs, EGP only requires 5 additional samples to disambiguate labels for the entire dataset. The baseline method SoftenSG still cannot resolve the ambiguity, achieving $ACC_{\mathbf{z}}$ of $0.014 \pm 0.001$ and $ACC_y$ of $0.883 \pm 0.004$.

Table 6 shows another comparison with the best-performing prototype-based method from Andolfi & Giunchiglia (2025), in the *Kand-Logic* dataset pulled from the original paper. As with *MNAdd-EvenOdd*, this task is unlearnable without observing additional information, as the supervision comes from rules that treat all classes symmetrically (*e.g. are all shapes the same?*). ProtoNeSy requires a small number of $\mathbf{z}$ labels as prototypes to resolve ambiguous labels. To supplement supervision in EGP and baselines, we annotate 10 samples from the dataset with a different rule-set, obtaining an additional $y$ label for those samples. We consider this a less demanding form of supervision than directly labeling $\mathbf{z}$. We provide additional details in the appendix. Using this additional supervision is not sufficient for the baseline methods. In contrast, EGP is able to match the performance of the ProtoNeSy using this weaker form of supervision, as it is able to take advantage of the additional information in the new $y$ labels to rule out ambiguous solutions. VAEL and GEDI are still not able to mitigate the ambiguity, despite the additional supervision. In VAEL, the reconstruction term is likely focused on low level details (e.g. shape edges) rather than disambiguating the semantic signal. As before, GEDI does not take into account the consistency between $\mathbf{x}$ and $\mathbf{z}$, leading to poor performance.

## 4.6   Analysis of EGP effect on neurosymbolic learning

In this experiment, we examine how EGP helps alleviate ambiguity in neurosymbolic grounding in the *MNAdd-Half* dataset. Specifically, we examine two variants of EGP which correspond to two ways to optimize equation (6). The first is EGP-Direct, which corresponds to the method developed in Section 3, where we optimize the approximate reconstruction term directly alongside the main objective. The second

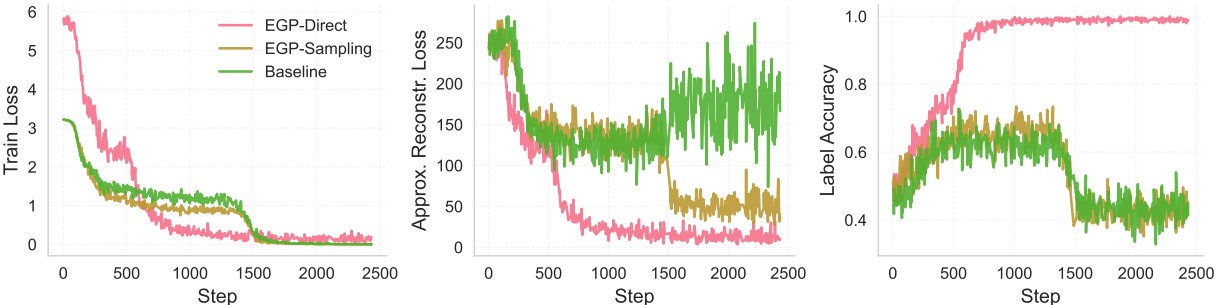

Figure 4: **Comparison of EGP variants on SoftenSG in *MNAdd-Half*.** The left panel shows the value of the training loss for each SoftenSG variant. The middle panel shows the value of the approximate reconstruction term and the right panel the inferred label accuracy. Label accuracy is directly related to the final grounding accuracy of the trained model. In both EGP-Direct and in the sampling-based variant, EGP does not interfere with optimizing the weak supervision loss. In the direct computataion case, it also improves grounding accuracy. In this dataset, EGP-Sampling is unable to resolve all RSes, as it only compares digits within a sequence, which is insufficient in this case.

is EGP-Sampling, which can be viewed as an approximate optimization of the ELBO in equation (5), as discussed in section 3.4. Here we study it more closely by combining it with SoftenSG, to facilitate a one-to-one comparison with EGP-Direct.

We focus on *MNAdd-Half* as it has two distinct incorrect solutions that aid in the analysis of EGP variants. Marconato et al. (2023) provide a detailed analysis but in short, one solution involves merging classes 2 and 4 and the other merges classes 1 and 3. The critical difference between the two is that samples of classes 2 and 4 appear in equations together (e.g. 2+4=6) whereas samples of 1 and 3 do not.

Figure 4 shows the training loss, the approximate reconstruction term and the grounding label accuracy over the three methods. In EGP-Direct, the training loss is the SoftenSG loss, which is a lower bound on equation (1), combined with our approximated reconstruction loss. In the other two cases it is exactly the SoftenSG lower bound. In all methods, there is an initial period of improvement where the model learns to satisfy some of the constraints. Then, improvement in the EGP-Direct variant is driven by the approximate reconstruction loss, as this continues to improve rapidly. This resolves all ambiguity as shown by the high inferred label accuracy. The only way for the model to both satisfy the constraint, and EGP-Direct, is to converge to the correct grounding. In contrast, the other two methods eventually converge to incorrect groundings, but their approximate reconstruction error deviates significantly.

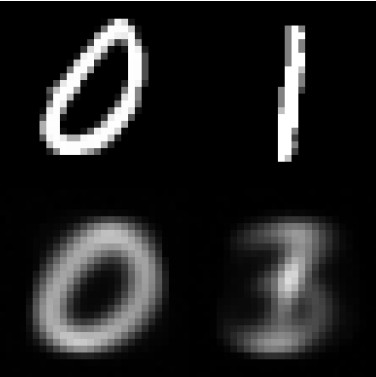

Figure 5: Reconstruction of MNIST digits after training on *MNAdd-Half*. The decoder can reduce reconstruction error by producing ambiguous and fuzzy images, failing to constraint the neurosymbolic model from merging underlying concept classes.

Closer examination of the trained models shows that EGP-Sampling consistently fails due to the second incorrect solution, where digit classes 1 and 3 are merged. This is because digits of these classes never appear together in a sample and as such are never forced to have different labels under EGP-Sampling. The second incorrect solution is eliminated however, which explains the lower approximate reconstruction error in figure 4. Interestingly, SoftenSG with EGP-Sampling also underperforms NeSyDM, which should fall victim to the same shortcut but performs better in our previous experiment, though still not as the direct approach. This is likely due to NeSyDM modeling the joint distribution of symbols and having to distribute

probability among fewer combinations. The baseline method consistently falls into one of the two failure modes, and its approximate reconstruction error increases rapidly, after the initial phase.

Finally, we use *MNAdd-Half* to provide additional insight on the result by Marconato et al. (2023) showing that pixel-level reconstruction is insufficient despite directly optimizing the reconstruction term in equation (5). In their experiment, they introduce an additional prediction head on the classifier which predicts a latent variable meant to capture stylistic information. We suspect this setup runs the risk of circumventing the neurosymbolic learner and carrying all information for reconstruction through the additional variable, thus eliminating any pressure on the grounding model to resolve the ambiguity. Therefore, we create an alternative setup where the decoder needs to learn to reconstruct digits directly from the predicted concepts, rather than having an alternate route for information. We use the gumbel softmax trick to propagate the reconstruction loss gradient to the encoder, which we train for grounding using SoftenSG. Additional details are available in the appendix. Still, using direct estimation of the reconstruction loss in equation (5) fails to provide perceptual grounding, achieving $ACC_y$ of $0.939 \pm 0.013$ and $ACC_{\mathbf{z}}$ of $0.435 \pm 0.007$, despite the reconstruction loss steadily decreasing and converging to a small value. Figure 5 provides some insight into why. The decoder is able to cheat by producing smeared out, faint digits that seem to balance the reconstruction loss due to an incorrect digit reconstruction, with loss due to low-level details. As a result, the encoder does not face significant pressure to resolve the ambiguity, and instead still merges classes 1 and 3 into the symbol 1, and permutes symbols 2 and 4, satisfying the reasoning constraints while achieving low reconstruction error. In contrast, our approximated reconstruction error in EGP focuses on semantic information by first learning the inherent cluster structure of the input, and then forcing the grounding model to be consistent with it. This provides an inductive bias that mitigates ambiguity in the weak supervision.

## 5 Conclusion

In this work, we addressed the issue of insufficient supervision in neurosymbolic learning resulting in reasoning shortcuts, where grounding models learn unintended mappings that satisfy symbolic constraints but fail to capture correct underlying semantics due to insufficient supervision.

To overcome this, we introduced Efficient Generative Proxies, a framework that approaches neurosymbolic learning from a generative perspective without the pitfalls of pixel-by-pixel generation. By leveraging a lightweight approximation of the generative likelihood, EGP provides perceptual grounding by enforcing consistency, ensuring that semantically similar inputs yield similar symbolic predictions and distinct inputs yield distinct predictions. This approach provides the benefits of the generative approach, namely using the inherent structure of the input data to resolve ambiguities in supervision, without requiring the reconstruction of low-level details that can obfuscate the semantic signal. Furthermore, our framework naturally incorporates typical neurosymbolic learning approaches within it. Our extensive experiments validate that integrating EGP into state-of-the-art neurosymbolic methods significantly improves concept grounding accuracy and effectively mitigates reasoning shortcuts across challenging tasks from the *RSbench* benchmark and a custom multi-concept ObjectMath dataset. Furthermore, we demonstrated that EGP is highly adaptable, allowing for the incorporation of domain-specific inductive biases to further enhance grounding performance in complex, multi-attribute environments. Finally, the EGP framework is flexible, providing a wide design space for perceptually grounding neurosymbolic learning models.

Despite the strong empirical results, EGP does have some limitations. Most importantly, EGP depends on pretrained encoders that preserve the semantic structure of the data. This can be reasonably easy to obtain when concepts dominate the input (e.g. MNIST digits). In more complex settings, this may not always be possible, and our theoretical analysis indicates that in that setting, some unintended mappings may persist. However, EGP can still provide an advantage in practice by enabling the injection of additional inductive biases, which can help separate concepts without the need for labeled data.

Finally, future work can extend EGP to yield similar benefits in other settings, such as Concept-Based Models (CBMs) (Bortolotti et al., 2025), which also suffer from reasoning shortcuts in many cases. One promising direction is to compare the identifiable VAE (iVAE) (Khemakhem et al., 2020) to the reconstruction term in EGP. If the reconstruction term can help identify a single correct explanation of the input data and the reasoning result, then it will direct the grounding model to learn informative and generalizable concepts.

**Acknowledgments**   The authors gratefully acknowledge the Action Editor and anonymous reviewers for their rigorous and constructive feedback. Their insightful suggestions, particularly in suggesting additional experiments and baselines, as well as in improving the clarity of the writing, were invaluable in refining and strengthening this work. In addition, Li-Ping Liu's work was supported by NSF Award 2239869.

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

## A    Proof of Proposition 1

*Proof.* Following the framework of Marconato et al. (2023), the number of deterministic maps $\alpha \in \mathcal{A}$ that minimize the standard neurosymbolic learning objective is given by the maps that satisfy the logical consistency constraint:

$$\bigwedge_{g \in \text{supp}(G)} \beta_K(\alpha(g)) = \beta_K(g) \tag{9}$$

Our proposed Efficient Generative Proxies (EGP) framework introduces an additional reconstruction term. Under the Ideal Disentangled Clustering assumption, the unsupervised representation perfectly recovers the equality of ground-truth concepts for each component $j$:

$$p(C_{ij}^{(j)} = 1 \mid x_i, x_j) \approx 1 \iff g_i^{(j)} = g_j^{(j)} \tag{10}$$

Maximizing the EGP term requires the learned mapping $\alpha$ to respect this clustering structure. For any two inputs $x_i, x_j$ generated by distinct ground-truth concepts $g_i^{(j)} \neq g_j^{(j)}$, the generative proxy forces the learned symbols to be distinct:

$$g_i^{(j)} \neq g_j^{(j)} \implies \alpha_j(g_i^{(j)}) \neq \alpha_j(g_j^{(j)}) \tag{11}$$

This imposes an injectivity constraint on each component-wise mapping $\alpha_j$. Therefore, a deterministic map $\alpha$ is optimal under our augmented objective if and only if it satisfies both the logical consistency constraint and the injectivity constraint for all components $j = 1, \ldots, k$.

The total number of optimal maps $N_{opt}$ is thus the sum over all possible maps $\alpha \in \mathcal{A}$ of the indicator function that both conditions hold:

$$N_{opt} = \sum_{\alpha \in \mathcal{A}} \mathbb{I} \left\{ \bigwedge_{g \in \text{supp}(G)} \beta_K(\alpha(g)) = \beta_K(g) \right\} \cdot \prod_{j=1}^{k} \mathbb{I} \left\{ \bigwedge_{\substack{g, g' \in \text{supp}(G) \\ g_j \neq g'_j}} \alpha_j(g_j) \neq \alpha_j(g'_j) \right\} \tag{12}$$

$\square$

## B    Additional Experiment Details and Results

### B.1    Varying EGP encoder

We use *MNAdd-Half* to validate EGP's sensitivity to the underlying encoder architecture. While our main experiments are derived using N2D (McConville et al., 2019), as it is specially designed to obtain clusterable embeddings, we repeat the experiment using a standard autoencoder and SimCLR (Chen et al., 2020). We run these EGP variants with SoftenSG. We find that in our experiment, EGP is robust to the choice of encoder.

| Metric | SoftenSG+EGP(AE) | SoftenSG+EGP(SimCLR) |
|---|---|---|
| $ACC_y$ | $0.944 \pm 0.001$ | $0.940 \pm 0.005$ |
| $ACC_c$ | $0.971 \pm 0.001$ | $0.970 \pm 0.002$ |
| $ACC_{y,ood1}$ | $0.938 \pm 0.004$ | $0.939 \pm 0.005$ |
| $ACC_{c,ood1}$ | $0.969 \pm 0.002$ | $0.969 \pm 0.003$ |

Table 7: *MNAdd-Half* results with varying encoder method

## B.2 Modifying NeSyDM with EGP

NeSyDM (van Krieken et al., 2025a) learns an expressive distribution over possible solutions to reflect the uncertainty due to reasoning shortcuts. Since it does not use the independence assumption, we are not able to directly apply EGP to the loss. Instead, we modify the variational posterior, responsible for sampling approximate labels, so that they satisfy the EGP. This indirectly optimizes EGP, as the model learns to place probability in symbol configurations that are consistent with the input structure. Adapting to our own notation, the NeSyDM posterior is: $p(\mathbf{z}^0|\mathbf{z}^t, \mathbf{x}, y) \propto p(\mathbf{z}^0|\mathbf{z}^t, \mathbf{x})r(\mathbf{z}^0|y)$, where $r_\beta(\tilde{\mathbf{z}}^0 \mid \mathbf{y}^0) \propto \exp\left(-\beta \sum_{i=1}^{Y} 1[\varphi(\tilde{\mathbf{z}}^0)_i \neq y_i^0]\right)$. Note that here the super-scripts correspond to the diffusion time dimension. NeSyDM uses large $\beta$, essentially sampling $K$ samples from the model and keeping the ones that satisfy the most constraints. We modify the posterior by incorporating the EGP in the energy function of $r$. This results in $r_{\beta,\gamma}(\tilde{\mathbf{z}}^0 \mid \mathbf{y}^0, \mathbf{x}) \propto \exp\left(-\beta \sum_{i=1}^{Y} 1[\varphi(\tilde{\mathbf{z}}^0)_i \neq y_i^0] - \gamma \log p(\mathbf{x}_0, \mathbf{x}_1|c_{01})\right)$. Selecting $\gamma$ is easy when $\beta$ is large, as the only terms that survive the first term satisfy the constraint. Then the second term only needs to select the ones that satisfy the EGP. We use $\gamma = 1$ in all our experiments. In our experiments, $\mathbf{x}$ only contains 2 images, so this term suffices. Alternatively we can compute the term across all pairs, or pick some at random.

## B.3 Analysis of Surrogate EGP.

*EGP-Posterior Matching (PM)* is a surrogate objective to equation (6) that aims to align the model's posterior $q_\theta(C_{i,j}|\mathbf{x}_i, \mathbf{x}_j)$ with that derived from the GMM. In our experiments performance between the two is similar in practice but EGP-PM may be simpler to implement and may interact better with some optimizers in some settings, as it is simply the binary cross entropy loss between the two distributions. The objective is:

$$L_{EGP-PM}(\mathbf{x}_i, \mathbf{x}_j) = BCE(q_\theta(c_{i,j}|\mathbf{x}_i, \mathbf{x}_j), p_{GMM}(c_{i,j}|\mathbf{x}_i, \mathbf{x}_j)). \tag{13}$$

The first distribution is derived from the classifier using the independence assumption, as in equation (6). The latter is the collision probability computed in closed form for the GMM.

Figure 4 shows two plots illustrating single runs of SoftenSG with different implementations of EGP applied to it. EGP-Direct refers to the direct implementation of the EGP term in equation (6). EGP-PM refers to the posterior-matching version of EGP, where the probability of two inputs belonging to the same concept assigned by the model is matched to that of the GMM. Finally, EGP-Sampling refers to modifying the label sampling procedure during weakly-supervised learning to sample labels that optimize the EGP loss. The baseline method is SoftenSG. Crucially, EGP-Sampling naturally only considers inputs in the same input sequence, as labels across samples are sampled independently.

As expected, EGP-Direct most directly minimizes the EGP loss. The surrogate EGP-PM loss behaves like an upper bound, as minimizing it ensures the EGP loss term remains low, while the opposite does not hold: Minimizing EGP-Direct eventually results to an increase in the EGP-PM loss value. This is because EGP-PM penalizes the model if it does not reflect the GMM's uncertainty over whether two samples belong in the same class, while EGP-Direct encourages high confidence regardless of the GMM's confidence. In short, we hypothesize that EGP-PM may enjoy improved training dynamics in some settings, as it uses the standard binary cross entropy loss and provides linear gradients to the model, whereas in EGP-Direct the gradient may vanish if the model is confidently wrong about two inputs belonging to the same concept.

## B.4 ObjectMath Dataset Details

The *ObjectMath* dataset is designed to evaluate multi-concept neurosymbolic grounding in a highly ambiguous setting. Each image in the dataset contains three distinct objects. Every object is characterized by three attributes: color (red, green, or blue), shape (cube, sphere, or cylinder), and texture (smooth or rough).

In the reasoning task, each possible shape and color is mapped to a specific integer value. The final label $y$ for a scene is computed by multiplying the attribute values of each object and then summing these products across all three objects. For example, if an object has a color mapped to $v_c$ and a shape mapped to $v_s$, its contribution to the sum is $v_c \times v_s$. The overall equation is $y = \sum_{i=1}^{3}(v_{c,i} \times v_{s,i})$.

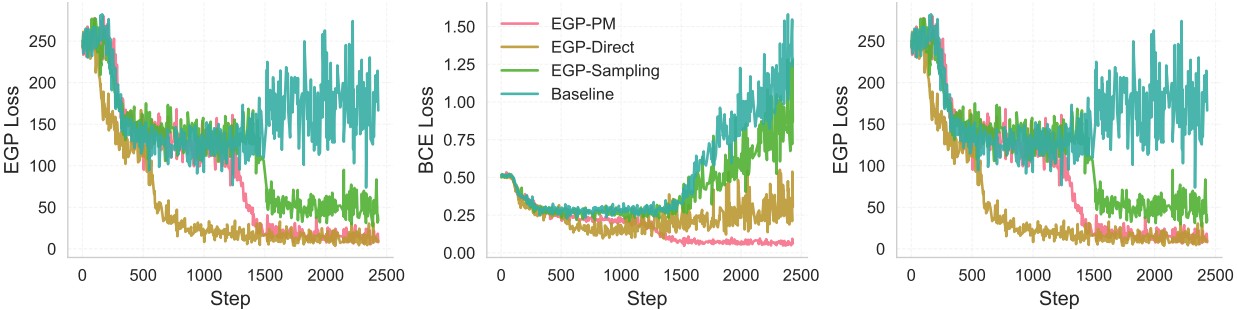

Figure 6: **Comparison of EGP variants on SoftenSG in *MNAdd-Half*.** Each panel shows the value of the EGP loss, the posterior matching BCE loss and the inferred label accuracy for different implementations of EGP during training. Label accuracy (right) refers to the quality of the labels sampled during inference and relates to the final performance of the trained model. EGP-Direct and EGP-PM both achieve high performance because they enforce prediction consistency across each training batch. Minimizing EGP-PM has a similar effect to minimizing EGP-Direct. EGP-Sampling indirectly minimizes the EGP loss as well, but only considers consistency within a sequence, which is insufficient in the MNAdd-Half dataset. Effective minimization of EGP-Direct is consistently associated with high quality labels.

This task is highly ambiguous for several reasons: 1. **Commutativity of Addition:** The order of the objects does not affect the final sum, meaning the model must learn to ground attributes consistently regardless of object position. 2. **Multiple Valid Combinations:** Many different combinations of attribute values can result in the same final sum, creating numerous reasoning shortcuts where the model could assign incorrect values to attributes while still perfectly predicting the label $y$.

To process this dataset, we first employ a standard object detection workflow to isolate the individual objects from the scene, yielding three separate image crops per scene. These crops are then passed to our neurosymbolic models.

### B.5    Additional Experiment Details

**Early stopping NeSyDM.**    We notice in the original NeSyDM experiments that grounding accuracy peaks when the model reaches stable predictions (predicted groundings on the validation set change slowly over training epochs). As such in our experiments we perform early stopping in all variants of NeSyDM by early stopping based on the minimum fraction of validation samples that change labels in each epoch, smoothed over 5 epochs. Crucially, this does not require validation grounding labels.

**MNIST EGP experiment settings.**    We implement EGP with a convolutional autoencoder for feature extraction and UMAP dimensionality reduction, following the N2D method (McConville et al., 2019). We then fit a Gaussian Mixture Model (GMM) for EGP on the resulting latent space. The autoencoder architecture consists of an encoder with three convolutional layers (channels: $1 \rightarrow 16 \rightarrow 32 \rightarrow 64$) followed by a linear layer projecting to a 10-dimensional embedding space, and a symmetric decoder using transposed convolutions. The model was trained on the MNIST training set for 10 epochs using the Adam optimizer with learning rate $10^{-3}$ and mean squared error reconstruction loss, with a batch size of 256.

Following training, embeddings were extracted from all training samples (batch size 1000) and reduced to 5 dimensions using UMAP with $n_{\text{neighbors}} = 30$, min_dist $= 0.0$ (to promote compact clusters) and cosine distance metric. A GMM with 10 components and full covariance matrices was then fitted to the UMAP-reduced embeddings. The resulting model computes cluster assignment probabilities and joint log-densities for image pairs under same-cluster and different-cluster hypotheses.

**Disentangled EGP Experimental Settings for ObjectMath**    .

The disentangled EGP method extends the standard EGP pipeline with a dual-encoder architecture that produces two separate, disentangled embeddings ($z_1$ and $z_2$) from a shared convolutional base. The architecture processes $64{\times}64$ images through a shared encoder (channels: $C \to 16 \to 32 \to 64 \to 128$) followed by two separate linear heads producing unit-normalized embeddings of dimensions $d_1$ and $d_2$. The decoder reconstructs from the concatenated embedding $[z_1, z_2]$ using transposed convolutions.

Training employs a two-stage adversarial objective using the ObjectMath training set. The reconstruction loss (weight $\lambda_{\text{recon}} = 1.0$) can be either pixel-wise MSE or a histogram-based loss that compares sorted pixel values, ignoring spatial structure to focus on color distribution. The disentanglement loss (weight $\lambda_{\text{disent}} = 0.1$) is implemented via cross-prediction MLPs that attempt to predict one encoding from the other. The MLPs are trained to minimize prediction error, while the encoders are trained to maximize it (via a negative term in the loss), encouraging independence between $z_1$ and $z_2$. The total loss is $\mathcal{L} = \lambda_{\text{recon}}\mathcal{L}_{\text{recon}} - \lambda_{\text{disent}}\mathcal{L}_{\text{disent}}$. Training uses the Adam optimizer with learning rates $10^{-3}$ for the autoencoder and $10^{-4}$ for the cross-prediction MLPs, with a batch size of 256 for 10 epochs.

Following training, separate UMAP reducers and GMMs are fitted to each embedding stream. Both UMAP models use $n_{\text{neighbors}} = 30$, min_dist $= 0.0$, $n_{\text{components}} = 5$, cosine metric. Each GMM uses $n_{\text{components}} = 3$ with full covariance matrices. The resulting model computes cluster probabilities and joint log-densities independently for each embedding, enabling analysis of different semantic dimensions captured by $z_1$ and $z_2$.

**Kand-Logic experiment additional rule set.** We use a simple synthetic rule to provide 1 bit of supervision to the 10 samples. The rule checks if for any of the objects in the scene, $z_1 < z_2$, meaning the class index of its shape is less than that of its color. We use the same approach as in the ObjectMath dataset to obtain disentangled encoders.

**EGP\* implementation for ObjectMath** Here we use a convolutional autoencoder with an encoder that progressively downsamples from $64{\times}64$ to $1{\times}1$ through four convolutional layers (channels: $C \to 16 \to 32 \to 64 \to 128$), followed by a linear projection to an embedding dimension. The decoder uses transposed convolutions to reconstruct the original image size, with a sigmoid activation for output normalization.

Training employs the Adam optimizer with learning rate $10^{-3}$ for 10 epochs with batch size 256. The reconstruction loss can be either standard pixel-wise MSE or a histogram-based loss that compares sorted pixel values across channels, effectively ignoring spatial structure to focus on color distribution. For the shape encoder we use regular image reconstruction over grayscale images instead. These loss variants enable training models that emphasize different aspects of the data (e.g., color-only or shape-only representations).

Following training, embeddings are extracted and reduced to 5 dimensions using UMAP with $n_{\text{neighbors}} = 30$, min_dist $= 0.0$, cosine metric, and random seed 42. A GMM with 3 components and full covariance matrices is fitted to the UMAP-reduced embeddings.

## B.6 Reconstruction experiment details

Here the model is a VAE with a discrete latent space that processes sequences of images using the Gumbel-Softmax trick for differentiable discrete sampling. The model consists of an encoder that maps individual images to hidden representations, a predictor that outputs logits for discrete latents (one per image in the sequence), and a decoder that reconstructs images from sampled discrete latents.

Training employs a multi-objective loss combining reconstruction and the semantic loss. The reconstruction loss uses mean squared error between original and reconstructed images, where reconstructions are obtained by encoding images to logits, sampling discrete latents via soft Gumbel-Softmax (temperature $T_{\text{gumbel}}$), and decoding.

Training uses the Adam optimizer with learning rate $10^{-4}$. Gumbel temperature undergoes exponential annealing from 1 to 0.1 with rate 0.99.

