# OpenReview forum: "Towards Overcoming Reasoning Shortcuts in Neurosymbolic Learning via Efficient Generative Proxies"
_TMLR — Accepted by TMLR_

### Review · Reviewer_pwwq · 2026-04-07

**Summary Of Contributions:**

This paper introduces Efficient Generative Proxies (EGPs) to implement a reconstruction regularization on Neuro-Symbolic (NeSy) models.
The rationale behind method introduction is to reduce or avoid reasoning shortcuts (RSs) in reasoning tasks enforcing a distinction between input with different latent concepts. The method leverages an approximation of the ELBO by modelling a similarity coefficient between input instances. Overall, the experimental verification on synthetic datasets shows an increase in concept grounding with EGP when applied to a variety of NeSy models.

**Strengths**.  The idea to study RSs and reconstruction with variational inference setting is valid and can give new ideas to progress on learning correct concepts. In particular, this may be helpful to connect identifiability theory with non-linear ICA [Khemakem et al. 2020].
1. The idea to repurpose generative models and discriminative classifiers to ground intermediate symbols is valid. The framework builds on variational inference and introduce an approximation better fit to avoid unstable trade-offs that are introduced in previous work [Marconato et al. 2023].  With EGP, reconstructing the input is not the main objective, the framework supports modelling a clusering, binary variable $C_{ij}$ that   tells when two inputs are considered equal by considering prototypes.
2. Theoretically, this gives an equivalent condition to the original reconstruction term used in [Marconato et al. 2023] that relaxes a content-style separation assumption. The new "clustering" condition can be easier to satisfy.
3. EGP offers a benefit in concept accuracy across datasets when compared to some unsupervised baselines. Moreover, I can see some improvements to standard reconstruction penalties, although this is not deeply investigated.
4. Finally, I appreciate some useful intuitions behind positive and negative results that are discussed in Section 4.3 and 4.4 which is very transparent of benefits and limitations of EGP in carefully chosen synthetic settings.

**Weaknesses**. With that being said, the paper requires substantial improvements before acceptance.
1. The paper does not give a convincing enough motivation behind the problem, specifically it builds on RSs works but lacks introducing the context of NeSy models and RSs. This can be done before section 3 by including examples of the problem they want to solve and how this is unaddressed. A discussion on the broader scope is also missing, narrowing the content to few methods in NeSy AI but missing the connection to identifiability in representation learning, see e.g. [Khemakem et al. 2020, Hyvarinen et al. 2024].
2. The authors do not consider natural competitor in the concept-unsupervised case. Models that consider generative priors and discriminative objectives have been proposed [Misino et al. 2022, Sansone and Manhaeve 2025] but they are not tested against EGP. Other unsupervised penalties include entropy maximization and contrastive learning [Marconato et al. 2025] that should be tested as well.
This does not give an accurate view on how EGP improves to other alternatives. Since [Misino et al. 2022] introduce a framework based on VAE, comparing it to EGP both from a model formulation and empirically is required.
3. Another weak point is that the experimental framework does not consider more realistic data. While results on MNIST and CLEVR are convincing, reconstruction penalties might be less fit in more realistic scenarios, where the input independent structure might not be present. Still, experimenting in that scenario and discussing which architectures are best fit to separate between object is relevant to understand the reach of EGP.
4. The proof of proposition 1 is missing in the appendix and this theoretical part in the appendix is not mentioned in the main text.
5. **Clarity of exposition can be improved**. The introduction of $C_{ij}$ is important but not well discussed in the paper.  The integer $b$ in 3.3 is not clear from the text, how big is it?  What approximations does EGP introduce in the original objective, and when is it supposed to work with RSs? How is $p(x | C)$ chosen? It is further unclear how authors choose to use pretrained modules to combine with EGP to make it work. This requires further explanations (see EGP* and the conclusion). \
Ablations of EGP in 4.3 are also not entirely clear, but it is said that they correspond to different choices of optimizing Equation 6. Can you further elaborate on that?  \
Figure 1 can be made more intuitive. If that is supposed to explain how EGP works to avoid RSs I think I could not get the intuition entirely. \
ObjectMath dataset is a bit unclear. I suggest having an apposite subsection in the appendix where the task is explained with some visuals about input data.

----

[Khemakem et al. 2020] Variational Autoencoders and Nonlinear ICA: A Unifying Framework, AISTATS \
[Hyvarinen et al. 2024]  Identifiability of latent-variable and structural-equation models: from linear to nonlinear, Annals of the Institute of Statistical Mathematics \
[Misino et al. 2022] VAEL: Bridging Variational Autoencoders and Probabilistic Logic Programming, NeurIPS \
[Sansone and Manhaeve, 2025] Unifying Self-Supervised Clustering and Energy-Based Models, TMLR

**Audience:**

Yes

**Audience Explanation:**

I give merit to the authors to have revisited reconstruction penalties and make them more effective for RSs. This is of interest for the community interested in symbol grounding in NeSy AI and explains how to combine unsupervised objectives with downstream tasks.
Limited evaluation however reduce impact of the proposed method.

The audience could be broader by investigating how EGP applies in cases where knowledge is not given beforehand but is learned as well, see [Bortolotti et al. 2025]. This in turns capture the setting of concept bottleneck models, that have been studied in a similar setting with reconstruction penalties [Alvarez-Mellis 2018, De Santis et al. 2025], or sparsity priors [Goyal et al. 2025].

----

[Bortolotti et al. 2025] Shortcuts and Identifiability in Concept-based Models from a Neuro-Symbolic Lens, NeurIPS \
[Alvarez-Mellis and Jaakkola, 2018] Towards Robust Interpretability with Self-Explaining Neural Networks, NeurIPS \
[De Santis et al., 2025] Towards Better Generalization and Interpretability in Unsupervised Concept-Based Models, ECML-PKDD \
[Goyal et al. 2025] Causal Differentiating Concepts: Interpreting LM Behavior via Causal Representation Learning, NeurIPS

**Broader Impact Concerns:**

The paper is very specific to NeSy models that use architectural biases to separate between distinct input parts, which limits the scope. Beyond that setting, the method is not readily applicable and can thus have a relatively small impact on the broader problem of how learning concepts without prior knowledge.

I suggest at least mentioning or, even better, experimenting with Concept Bottleneck Models in the setting described by [Bortolotti et al., 2025]. This allows to discuss EGP in relation to more challenging tasks that may support inductive biases such as input disentanglement but lag behind in promoting correct concept learning.

Even within NeSy community, the broader impact is limited. The idea of focusing on input (dis)similarities is helpful but can be discussed in relation to other methods that consider similar objectives. Clustering can be compared to self-supervised methods [Sansone and Manhaeve, 2025] and VAE implementations [Misino et al., 2022] in NeSy learning. The paper currently cites [Sansone and Manhaeve, 2025] but does not offer a comparison.

**Claims And Evidence:**

No

**Claims Explanation:**

Regarding whether EGP can promote better grounding of concepts when RSs are present is not supported by entirely convincing evidence.

- Guarantees from theory are unclear (see Appendix A). Assumption A.1 is introduced but it is not clear where it enters in the EGP modeling. Specifically, the proof of Proposition 1 is missing, which could highlight how Asm A.1 is used. Furthermore, it is not discussed when this can hold.
- The experimental evaluation only comprehends synthetic and ad-hoc settings where concepts can be easily separated.
- The derivation of the variational framework is not entirely novel, see [Misino et al. 2022], but this is a minor point.

**Requested Changes:**

To strengthen the paper, I suggest the following changes:
- **Integrating competitors** Experimenting against other concept-unsupervised mitigation strategies (entropy maximization, contrastive learning, VAEL [Misino et al. 2022]) is missing and should be included for all synthetic datasets.
- **Experiments on less synthetic settings**. Validating EGP is important in contexts where assumptions can be (marginally) violated. To validate whether the clustering variable facilitates correct concept grounding beyond other natural baselines, it should be investigated in more realistic settings. For example datasets that include rules and learnable concepts: BDD-OIA [Sawada and Nakamura, 2022] or SDD-OIA [Bortolotti et al., 2024], Pascal-VOC [Donadello et al. 2017], MonumAI [Lamas et al. 2021].
- **Including proofs of the statements** This is missing and should discussed in relation to [Marconato et al. 2023].
- **Improve clarity by resolving weaknesses above**.

To broaden the scope, I suggest experimenting with Concept Bottleneck Models [Bortolotti et al. 2025]:
- Same synthetic datasets, but without fixed prior knowledge. Especially, with linear layers implemented as in concept bottleneck models.

----
[Sawada and Nakamura, 2022] Concept Bottleneck Model with Additional Unsupervised Concepts, CVPR \
[Donadello et al. 2017] Logic Tensor Networks for Semantic Image Interpretation, IJCAI \
[Lamas et al. 2021] MonuMAI: Dataset, deep learning pipeline and citizen science based app for monumental heritage taxonomy and classification, Neurocomputing

---

> ### Author Response · Authors · 2026-04-28
> **Thank you for your feedback**
>
> We thank the reviewer for the encouraging comments about the probabilistic formulation and the reconstruction proxy. We also thank the reviewer for suggesting directions to improve the work. Below, we address the concerns.
>
> 1. *“The paper does not give a convincing enough motivation behind the problem” … “lacks introducing the context of NeSy models and RSs”, “missing the connection to identifiability in representation learning,”.*
>
> We will improve the writing as suggested. We will add an example and formulate the problem in a more rigorous NeSy way.
>
> We will also include a discussion of identifiability. In this work, we mainly consider the case where there are multiple z explanations that are consistent with y. The same issue is also discussed in Eq.(3) of [1]. By implicitly reconstructing the key information of the data, EGP eliminates reasoning shortcuts for a wide range of problems. Here the similarity calculation is the inductive bias, which is a well-studied problem in machine learning. For harder problems (Eq. 5 in [1]), multiple labelings of z can explain the data, so the true z labels cannot be inferred from the data, and more labeled data is needed [2].
>
> 2. *"The authors do not consider natural competitor in the concept-unsupervised case"*
>
>
> We added two experiments to compare EGP against the two baselines :
> VAEL = [Misino, 2022]
> GEDI = [Sansone and Manhaeve, 2025]
>
> Dataset: MNAdd-half
>
> | Model | ACC_z | ACC_y |
> | :--- | :--- | :--- |
> | VAEL | 0.45 ± 0.11 | 0.97 ± 0.12 |
> | GEDI | 0.60 ± 0.01 | 0.42 ± 0.01 |
>
> Dataset: MNAdd-EvenOdd
>
> | Model | ACC_z | ACC_y |
> | :--- | :--- | :--- |
> | VAEL | 0.07 ± 0.06 | 0.96 ± 0.09 |
> | GEDI | 0.34 ± 0.02 | 0.13 ± 0.01 |
>
> VAEL is close in principle to ours and prior works' reconstruction experiments, as it uses a decoder. It fails to resolve the ambiguity in these tasks because of low level pixel details.
> GEDI is able to mitigate some ambiguity: its objective specifically prevents extremely degenerate solutions, as shown in the original paper. However, in the NeSy setting, it still relies on the semantic loss to satisfy constraints and does not address the consistency between choices of z and x.
>
> 3. *"Another weak point is that the experimental framework does not consider more realistic data."*
>
> We use a new dataset, the MonumAI dataset, to test the proposed EGP model. We construct a NeSy task by constructing a rule set to map buildings into architectural styles (y) from their architectural elements (z). We filter out samples that do not conform to the rules. These rules are ambiguous to give weak supervision, but they preserve the correct building mapping in the filtered dataset.
>
> The dataset has bounding boxes of architectural elements. With the visual input (x) of an architectural element, the model is trained to predict the element label (z). All the supervision information is from the style label (y) and the rule set.
>
> We then train 3 variants: SoftenSG, SoftenSG+R (which reconstructs pixels) and SoftenSG+EGP, which combines our method with the baseline. We use pretrained ResNet features for the encoder of EGP.
>
> The results are:
>
> | Method | Avg. **z** F1 ↑ | Label F1 ↑ |
> | :--- | :--- | :--- |
> | SoftenSG | 57.32 ± 1.21 | 92.11 ± 2.41 |
> | SoftenSG+R | 56.87 ± 1.32 | 91.74 ± 2.39 |
> | SoftenSG+EGP | **90.26 ± 3.22** | 93.67 ± 2.15 |
>
> We report F1 due to class imbalance. We still see that EGP can significantly outperform the baseline method, though the visual input is much more complex than MNIST digits or objects in the CLEVER dataset.
>
> We agree that separating objects from the input is a challenging problem. One possible direction is to run an object detection model over the input and get objects in separate bounding boxes. We can use the object classification head as our predictive model and fine-tune it within our EGP framework. We plan to leave this piece of work for our next paper.
>
> 4. *"The proof of proposition 1 is missing in the appendix, and this theoretical part in the appendix is not mentioned in the main text."*
>
> We will move the proposition to the main text and provide the proof in the appendix.
>
>
> [1] Marconato, Emanuele, Stefano Teso, and Andrea Passerini. "Neuro-Symbolic Reasoning Shortcuts: Mitigation Strategies and their Limitations." 17th International Workshop on Neural-Symbolic Learning and Reasoning. 2023.
>
> [2] Luca Andolfi and Eleonora Giunchiglia. Right for the Right Reasons: Avoiding Reasoning Shortcuts via Prototypical Neurosymbolic AI. In: The Thirty-ninth Annual Conference on Neural Information Processing Systems. 2026.

---

> > ### Author Response · Authors · 2026-04-28
> > **continuation**
> >
> > 5. *"The introduction of  C_ij is important but not well discussed in the paper. The integer  in 3.3 is not clear from the text, how big is it? What approximations does EGP introduce in the original objective, and when is it supposed to work with RSs? How p(x | C) is chosen?"*
> >
> > We will revise the discussion to improve clarity for the $c_{ij}$ term. We develop the theory using one reasoning problem for the sake of simplicity of notations, which indicates that x inputs should only be compared within a reasoning problem.
> >
> > In practice, we compare instances within a training batch. In this regard, $b$ is related to the batch, as we compare pairs of z labels in the same batch, so $b = LK$, with $L$ being the number of digits (e.g. 2) and  $K$ being the batch size. This method allows more comparison on z labels and is a convenient technique to improve the performance.
> >
> > As in our response to 2ZPg, we can use a full probabilistic approach without approximation, then we will need to assign classes to clusters, which allow interactions between all instances. However, this approach would require extra “global” parameters in the model, which we want to avoid.
> > Our current approximation still keeps the essence of comparing the distance between instances, therefore, it still keeps the power to disambiguate z labels by their visual similarity.
> >
> > In the clustering formulation, $p(\mathbf{x} | \mathbf{z})$ and $p(\mathbf{x} | C)$ are the same probability. We derive the cluster structure from $\mathbf{z}$: the comparison between $z_i$ and $z_j$ indicates whether $x_i$ and $x_j$ are in the same cluster or not. Therefore, $C$ contains all information about the cluster structure, so we use $p(\mathbf{x} | C)$ without referring to the actual labels.
> >
> > 6. *“It is further unclear how authors choose to use pretrained modules to combine with EGP to make it work. This requires further explanations”*.
> >
> > We tried a few pre-trained representation modules, and they performed similarly (B.1 in they appendix). In the problems investigated in our experiments, a typical representation learning module seems to be able to cluster visual inputs well.
> >
> > 7. *“Ablations of EGP in 4.3 are also not entirely clear, but it is said that they correspond to different choices of optimizing Equation 6. Can you further elaborate on that?”*
> >
> > In this ablation study, EGP-direct is the method developed in section 3. Baseline is the SoftenSG. EGP-sampling can be viewed as an approximate optimization of the ELBO in (5). Direct optimization of equation (5) encourages z samples from the cluster structure because of the reconstruction term. In EGP-sampling, we directly derive weights of z samples from the cluster structure and use them to modify the sampling procedure in the first term. The sampling procedure is from the baseline algorithm (SoftenSG in 4.3 or NeSyDM in other experiments).
> >
> > In this ablation study here, we want to show that EGP-sampling can still improve the second term over the optimization procedure, but not as well as EGP-direct. However, the baseline method might reach solutions violating the clustering structure.
> >
> > The ablation study confirms that EGP-sampling is a reasonable approximation when it is combined with NeSyDM, which has a complex sampling procedure and does not allow easy computation of the second term of (5).
> >
> >
> > 8. *“Figure 1 can be made more intuitive.”*
> >
> >
> > We will update Figure 1 in the next version.
> >
> > 9. *“ObjectMath dataset is a bit unclear.”*
> >
> > We will add more details and plots about this dataset.

---

> ### Author Response · Authors · 2026-05-05
>
> Thank you once again for your feedback on our submission. Please let us know if you have further questions as we revise our manuscript and based on our previous responses.

---

> > ### Comment · Reviewer_pwwq · 2026-05-13
> > **Reply to authors**
> >
> > I thank the authors for the lengthy response and addressing some points I have raised. I also double-checked some sections I flagged in my review.
> >
> > I summarize the points that were subject to modifications:
> >
> > > Connection to works in identifiability of generative models and reasoning shortcuts in CBMs
> >
> > The connection to works in identifiability of representations (e.g., iVAEs) should be included. The discussion on future work on CBM is plausible, but in that there could be a further connection to identifiability of generative models, as in iVAEs.
> >
> > > Inclusion of the competitors GEDI, VAEL, and Protopnet
> >
> > The comparison is very useful and improvements are visible. However, why did the authors experiment only on MNIST-Addition variants for GEDI and VAEL and why did they choose KAND for Protopnet? Why protopnet has zero standard deviation? This should be clear in the experimental section outline (see also clarity below).
> >
> > > Experiments on MonumAI
> >
> > This analysis is valuabe and shows good performance of EGP. Why did the authors experiment only with SoftenSG?
> >
> > > Missing theory
> >
> > This part improved. Though, before the proof, assumptions A1 and A2 should be explicitly stated.
> >
> > > Clarity
> >
> > I still believe the presentation can improve in clarity.
> >
> > - The introduction of RSs can be further improved (do the authors have an example in mind that can be solved with EGP, for example in MNIST-Half?).
> >
> > - Discussion on $C_{ij}$ is improved, although from the discussion and what is written in text, it is not clear how p(x | c) trades-off reconstruction quality w.r.t. p(x | z). In this respect, how can $C_{ij}$ capture different stylistic cues of the latent representations?
> >
> > - The scalar $b$ should also be clearly stated in the text.
> >
> > - Ablation. Thank you for the clarification.
> >
> > - **Experiments**. I do not understand the rationale of many parts of the empirical verification. This should be improved, e.g., collecting methods and datasets in a separate section, clarifying **why some methods are tested on some benchmarks**, and the choices for comparison in the ablation. Right now, the experiments section is really scattered and fails to convey a precise message and the choice of the comparison between methods. Restructuring this section could help sharpen the message and highlight the key takeaways.
> >
> > > Are the claims made in the submission supported by accurate, convincing and clear evidence?
> >
> > I would like to further discuss this point. Because EGP does not always offer "provable" guarantees of shortcut removal, claiming that RSs are addressed by EGP is incorrect. The correct claim is that EGP mitigates some RSs and improves concept grounding (which is correctly mentioned in some parts of the paper).
> >
> > I **strongly** advise the authors to resolve this, and avoid terms as "overcoming/solving" referred to EGP (in the title, introduction, conclusion) and turn it to "mitigating".

---

> > > ### Author Response · Authors · 2026-05-14
> > > **Thank you for further feedback**
> > >
> > > Here is our plan addressing your concerns.
> > >
> > > 1. We will further improve our writing. In particular, we agree that "mitigating" is a more accurate than "vercoming/solving". We will change the title and also revise relevant descriptions in the text.
> > > 2. We will add more experiment comparisons and restructure the experiment section. Previously, we wanted to provide quick comparisons, so newly added datasets do not include all algorithms. Now we will include four baseline algorithms on all datasets. We will also restructure the writing of the experiment section.
> > >
> > >  It is hard to extend NeSyDM or Prototypical NeSy to datasets beyond those in the original paper. On three datasets, NeSyDM + EGP has already outperformed NeSyDM. EGP also has an advantage over Prototypical NeSy because it does not require labeled instances. We hope this plan is acceptable.
> > >
> > > We will upload the next version very soon to include all the proposed changes.
> > >
> > > Thank you again for sharing your insights.

---

> > > > ### Author Response · Authors · 2026-05-15
> > > > **Uploaded a new revision**
> > > >
> > > > We have updated our manuscript with additional baselines across experiments. We have also revised the writing and structure, particularly of the experiments, to make them more coherent. We also changed the title and modified the text away from "resolving" shortcuts.
> > > >
> > > > Once again thank you for your feedback. Please let us know if there are any further questions or concerns.

---

> > > ### Author Response · Authors · 2026-05-15
> > > **Detailed responses to requested changes**
> > >
> > > 1. *The connection to works in identifiability of representations (e.g., iVAEs) should be included.*
> > >
> > > We understand the work in the following way. Grounding subsymbolic inputs is not identifiable if it is only done with reasoning with the observed label $y$. However, iVAE provides an identifiable result with continuous random variables. Therefore, combining iVAE and reasoning might provide a method to address the issue of reasoning shortcuts.
> > >
> > > 2. *Inclusion of the competitors GEDI, VAEL, and Protopnet*
> > >
> > > We have added experiments to include GEDI and VAEL in all datasets. EGP still outperforms the two methods on all datasets except MNLogic. On the MNLogic dataset, the two methods eliminate ambiguity and perform almost perfectly, matching the performance of EGP.
> > >
> > > We separate the comparison with Protopnet (named ProtoNeSy in the submission) in section 4.4, where extra supervision information is provided. When the supervision information is very weak, there are reasoning shortcuts that EGP cannot resolve. However, with only a few extra reasoning instances, EGP can achieve a similar performance to ProtoNeSy. We have the details in section 4.5.
> > >
> > > We have restructured the entire experiment section. We hope that this section is easier to read.
> > >
> > > 3. *before the proof, assumptions A1 and A2 should be explicitly stated.*
> > >
> > > We have explicitly made the assumption clear before the proposition.
> > >
> > > 4. *The introduction of RSs can be further improved ... an example can be solved with EGP*
> > >
> > > Our updated main figure has an example. We put a discussion after the problem definition (immediately before section 3.1).
> > >
> > > 5. *it is not clear how p(x | c) trades-off reconstruction quality w.r.t. p(x | z). In this respect, how can
> > >  capture different stylistic cues of the latent representations?*
> > >
> > > We expanded our discussion in 3.3. In our work, we hypothesize that the stylistic cue is not relevant to the main task, so $z$ should not reconstruct such low-level details. The cluster relationship $C$ blocks such information, so $z$ only needs to reconstruct $C$. In the inference, being in the same cluster or not helps to disambiguate $z$.
> > >
> > > 6. *The scalar b should also be clearly stated in the text.*
> > >
> > > We realize that we should use $n$, the number of input instances in a reasoning problem. Later, we expand the comparison to be between any pair of input instances in the dataset. We have improved our discussion in 3.3.
> > >
> > > 7. *Because EGP does not always offer "provable" guarantees of shortcut removal, claiming that RSs are addressed by EGP is incorrect.*
> > >
> > > We may have unconscious overclaim. We have updated our title to be "Towards Overcoming Reasoning Shortcuts ..." to indicate that our work is one step toward the goal. We have also changed phrases that contain "overcome" and "solve" to make the claim proper. We hope this change addresses your concern.

---

### Review · Reviewer_2ZPg · 2026-04-13

**Summary Of Contributions:**

This paper proposes Efficient Generative Proxies (EGP), a regularization objective to address reasoning-shortcuts, where grounding models learn unintended input-to-symbol mappings that satisfy output constraints but fail to capture correct semantics. The paper illustrates reasoning shortcuts with an example of MNIST digit addition, where a model can merge digit classes (e.g., collapsing classes 0 and 2 into a single symbol) while still satisfying all sum constraints, achieving high label accuracy on without correctly grounding the individual digit symbols. tuitively, the EGP regularization term is focused on flagging cases where the classifier assigns the same symbol to inputs that a separately trained unsupervised encoder places in different clusters, or different symbols to inputs in the same cluster — thereby penalizing the class-merging behavior that characterizes reasoning shortcuts. The method is demonstrated as a plug-in for existing neurosymbolic methods (SoftenSG, A2G, NeSyDM) on the RSbench benchmark and a custom multi-concept ObjectMath dataset.

Strengths:
1. The core idea is simple and elegant. I also appreciated the theoretical analysis in Proposition 1 showing that EGP reduces the space of valid groundings.
2. The regularization term shows significant improvements in all tested cases.

**Audience:**

Yes

**Audience Explanation:**

This is an interesting approach to mitigate reasoning-shortcuts without requiring high-quality pixel-level reconstruction.

**Broader Impact Concerns:**

NA.

**Claims And Evidence:**

No

**Claims Explanation:**

While the claims in the paper are technically substantiated by the evidence, I feel that the paper is missing a simpler yet potentially powerful baseline that could effective solve the problem at hand without the need for a neurosymbolic framework. The unsupervised encoder + GMM pipeline appears to already solve the grounding problem on its own. For example, if the GMM with 10 components cleanly separates MNIST digit classes (which it must, given 98%+ accuracy with EGP), can one not skip the entire neurosymbolic framework and directly predict z from the unsupervised encoder? Thus, the central claim that EGP resolves reasoning shortcuts through its generative framework is not convincingly disentangled from a simpler explanation: the unsupervised encoder already captures the cluster structure well enough to predict z directly.

**Requested Changes:**

I request the authors to demonstrate how the neurosymbolic learning framework adds value beyond just utilizing the unsupervised pretrained encoder.

Another limitation of this work is that the pairwise reconstruction term fundamentally requires n $\ge$ 2 instances per training example. This limits the method's applicability to multi-instance settings where a symbolic constraint aggregates over multiple inputs. If the authors agree, it would be good to add a discussion around this practical limitation in the paper.

---

> ### Author Response · Authors · 2026-04-28
>
> Thank you for the positive feedback about the core idea.
>
> 1. *"The unsupervised encoder already captures the cluster structure well enough to predict z directly."*
>
> In neural-symbolic learning, the logic reasoning part provides labels for the model to ground visual inputs to symbols. This is not feasible from a pure unsupervised method, including clustering. In this MNIST case, the clustering method can discover the structure but cannot give labels.
>
> Furthermore, clustering is just one way to reconstruct relevant information in the input. In the last part of 3.3, we extend the method to help the model learn object properties such as shapes and colors. It is hard to obtain discrete labels for these properties without the reasoning part.
>
> 2. *"Another limitation of this work is that the pairwise reconstruction term fundamentally requires $n\geq2$ instances per training example."*
>
> We appreciate your insight here. We understand this issue from theory and practice. On the theory side, there is an inference method that can actually consider the label assignment for all instances. In Eq. 6, we approximate $p(x | z)$ with $p(x_i, x_j | c_{ij})$ terms. If we use the GMM in $p(x | z)$ without the approximation, we will need to assign a class label to each cluster according to the labels of instances in the cluster. The cluster label would be a global latent variable to be inferred. While it is still possible, this method would increase learnable parameters and make batch training harder. So we chose our current method.
>
> On the practice side, we actually compare $x$ inputs not only from the same reasoning problem but also inputs of all problems in the same batch. With batch training, all pairs of $x$-s have chances to be compared. Our experiment shows that this trick can improve model performance as it enables comparisons of inputs across samples.
> In the next version, we will include a more detailed discussion of this modeling choice and the training technique.

---

> ### Author Response · Authors · 2026-05-05
>
> Thank you again for your comments and feedback on our work. Please let us know if there are any further questions based on the latest revision or our previous comments.

---

### Review · Reviewer_YQPe · 2026-04-24

**Summary Of Contributions:**

The paper addresses the phenomenon of "reasoning shortcuts" in neurosymbolic AI approaches --- this is a common failure model. The approach that the authors take is via a new "grounding" perspective (that they call EGP). The method is related to a previous formulation by Marconato et al (2023) but with a generative interpretation. The high level idea is that perceptually similar inputs should yield similar symbolic predictions, and dissimilar inputs yield distinct predictions. The authors show several interesting results on challenging neurosymbolic benchmarks such as RSBench, MNAddHalf, and ObjectMath.

**Additional Comments:**

Dear AE, my apologies for the delay, and for the brief review. This is not in my area of expertise, so my understanding of the prior literature is limited. I am also an AE myself, and am swamped with many other reviews/reviewers...

**Audience:**

Yes

**Audience Explanation:**

Topic (neurosymbolic reasoning, generative models) is certainly in scope for the typical TMLR reader.

**Claims And Evidence:**

No

**Claims Explanation:**

Clarity: The paper is generally clearly written. I enjoyed reading it.

Accurate: The paper is technically sound, the theory seems solid, and the experimental results are comprehensive.

Convincing: Generally, okay -- but it would be great to have included a bit more discussion of the more recent literature. For example, the recent NeurIPS paper by Andolfi and Giunchiglia (2025) introduces Prototypical Neurosymbolic architectures, which uses prototypical learning (another idea borrowed from self-supervision) to avoid reasoning shortcuts. Is there a generative interpretation of such methods, and can the authors connect EGP to their work?

**Requested Changes:**

I will support publication if the authors can include some discussion of, and ideally some comparisons with Andolfi-Guinchiglia (2025).

---

> ### Author Response · Authors · 2026-04-27
> **Thank you for your comments**
>
> Thank you for your positive feedback that our work provides clear and accurate evidence to support the proposed new method.
>
> 1. About your comment: *it would be great to have included a bit more discussion of the more recent literature*
>
> Thank you for pointing us to the Prototypical NeSy method. The work also addresses the reasoning shortcut problem. This approach creates a centroid/prototype for each class and then maps instances in the same class to the same centroid. To create these centroids, the method requires *labeled instances* from some classes. The new method shows good performance on several datasets.
>
> Compared to Prototypical NeSy, our method shares a similar spirit of "using clustering" information from the data. However, our method differs from this work in the following two aspects:
> 1. EGP does not require z labels (C labels in [1]). All supervision of EGP model training is from the reasoning label y. The model leverages the internal data structure to help disambiguate latent labels z.
> 2. We develop our method from a generative approach with a rigorous probability calculation. From the formulation in Eq. (5), EGP re-writes the "recovery" term and incorporates the clustering and multi-view clustering information. It is relatively easier to derive further methods from Eq. (5).
>
> Despite that our model needs less supervision, we will still make a comparison to Prototypical NeSy and post results as soon as we have them. We will also include the discussion above in our next version.
>
> [1] Luca Andolfi and Eleonora Giunchiglia. Right for the Right Reasons: Avoiding Reasoning Shortcuts via Prototypical Neurosymbolic AI. In: The Thirty-ninth Annual Conference on Neural Information Processing Systems. 2026.

---

> > ### Author Response · Authors · 2026-04-28
> > **A separate note**
> >
> > As a separate note, our current version cited the Prototypical NeSy method [Andolfi and Giunchiglia], but we will improve the discussion.
> >
> > In our experiment, we evaluated our model on the MNIST-EvenOdd dataset. The results are in Table 2. Prototypical NeSy [Andolfi and Giunchiglia] was also evaluated on the same dataset in the original paper (Table 1).
> >
> > In this dataset, all reasoning problems (both x and y) can be explained by multiple z labelings, which means that the accurate inference of z labels is not possible. Therefore, we see that EGP achieved lower performance than Prototypical NeSy when EGP received no extra supervision.
> >
> > Nevertheless, in section 2.4, when we add slightly more supervision, EGP achieves grounding accuracy 0.913 ± 0.004,  which is slightly worse than the performance (0.95 to 0.98) of Prototypical NeSy. However, this additional supervision only comes from 5 more samples (0+5)=5, (1+6)=7, (2+7)=9, (3+8)=11, (4+9)=13, where digits in brackets are MNIST digits. These five samples help disambiguate $z$ labels of the entire dataset. We argue that our supervision form is more convenient than Prototypical NeSy because we may only need the practitioner to collect a few more samples, instead of directly labeling the visual inputs.
> >
> > The results are in the current version, but we will extend the discussion. We will work on the Kand-Logic dataset and get more comparisons to Prototypical NeSy.

---

> ### Author Response · Authors · 2026-05-05
>
> Once again, thank you for your feedback on our submission. Please let us know if you have any further questions and comments as we are working through revisions.

---

### Author Response · Authors · 2026-04-28
**We will update a new version soon**

Thank you all for your feedback and insights!

We have tried our best to respond to your questions. We will update our submission within the next few days to incorporate all of our promised revisions. We are eager to hear more from you!

--Authors

---

### Author Response · Authors · 2026-05-05
**Uploaded revised manuscript**

We have uploaded a new version of the manuscript, which contains most of our promised changes. We will post another revision with the following changes as soon as possible:

1. Experiments with VAEL and GEDI results with the OOD case. We have posted results on standard datasets. They perform poorly in the standard grounding case, so their performances are unlikely to be good in OOD cases.
2. We are working on the comparison against the Prototypical NeSy method on another dataset in its original paper.

--Authors

---

> ### Author Response · Authors · 2026-05-08
>
> We have uploaded a revised version with these experiment results.
>
> In the VAEL and GEDI cases, the performance in the OOD test sets follows the same pattern established in prior experiments: VAEL does not resolve the ambiguity as it relies on pixel-level supervision, and performs on-par with other baselines. GEDI performs better, as it can resolve some ambiguity, but not as well as our approach as it does not ensure consistency between choices of $\mathbf{z}$ and $\mathbf{x}$.
>
> We include an additional comparison with the Prototypical NeSy method on the Kand-Logic dataset which is used in the original paper. This dataset consists of rules that treat classes symmetrically (e.g. *are all shapes the same?*) and as a result is not learnable without additional supervision. The Prototypical NeSy method relies on some $\mathbf{z}$ labels to make the task learnable. In our experiment we are able to match the performance with a weaker form of supervision: We only label 10 samples with an additional $y$ label derived from a different reasoning task than the original dataset. We consider this a more convenient form of supervision as it does not rely on concept-level annotations. We provide additional details in the revised manuscript.
>
> Baseline approaches still fail in this setting, showing that there is still substantial ambiguity in the supervision signal, but EGP is able to resolve it by finding $\mathbf{z}$ that explain all the $y$ labels as well as the inputs $x$.
>
> Once again, we would like to thank all reviewers for their feedback and for helping strengthen our manuscript. We will be closely monitoring the forum in case of additional questions or concerns.
>
> --Authors

---

### Author Response · Authors · 2026-05-12

We have uploaded another revision of the manuscript, updating Figure 1 to better illustrate our approach.

Please let us know if there are any other questions or concerns.

Thank you!
-- Authors

---

### Decision · Action_Editor_VsZ1 · 2026-06-05

**Recommendation:** Accept as is

**Audience:**

Yes

**Audience Explanation:**

Neurosymbolic reasoning and, in particular, reconstruction methods.

**Claims And Evidence:**

Yes

**Claims Explanation:**

This paper looks at the problem of shorcuts in symbolic grounding. They propose a reconstruction-based training approach, showing improved performances on several established benchmarks such as RSbench and ObjectMap. These reconstruction methods already existed in neurosymbolic AI, but did not work as well. Concerns of reviewers were mostly addressed, one point was on how the neurosymbolic learning framework adds value beyond just utilizing the unsupervised pretrained encoder. The authros have responded but it would be good to also clarify this in the paper with a footnote or a sentence.

---

> ### Author Response · Authors · 2026-06-10
> **Thank reviewers for the feedback**
>
> We sincerely thank all reviewers for their thoughtful feedback. Your comments have helped us view our work from different perspectives and have contributed significantly to improving the paper. We also extend our gratitude to the Action Editor for guiding the review process and for your time and effort throughout. We will incorporate the final revisions and submit the camera-ready version shortly.
>
> --authors

---

> > ### Author Response · Authors · 2026-06-30
> > **We have posted the camera-ready version**
> >
> > We have posted the camera-ready version! Thank you again for the review work.
> >
> > --authors